# Inflationary Flows: Calibrated Bayesian Inference with Diffusion-Based Models

**Daniela de Albuquerque**
Department of Electrical &
Computer Engineering
School of Medicine
Duke University
Durham, NC 27708
daniela.de.albuquerque@duke.edu

**John Pearson**
Department of Neurobiology
Department of Electrical &
Computer Engineering
Center for Cognitive Neuroscience
Duke University
Durham, NC 27708
john.pearson@duke.edu

## Abstract

Beyond estimating parameters of interest from data, one of the key goals of statistical inference is to properly quantify uncertainty in these estimates. In Bayesian inference, this uncertainty is provided by the posterior distribution, the computation of which typically involves an intractable high-dimensional integral. Among available approximation methods, sampling-based approaches come with strong theoretical guarantees but scale poorly to large problems, while variational approaches scale well but offer few theoretical guarantees. In particular, variational methods are known to produce overconfident estimates of posterior uncertainty and are typically non-identifiable, with many latent variable configurations generating equivalent predictions. Here, we address these challenges by showing how diffusion-based models (DBMs), which have recently produced state-of-the-art performance in generative modeling tasks, can be repurposed for performing calibrated, identifiable Bayesian inference. By exploiting a previously established connection between the stochastic and probability flow ordinary differential equations (pfODEs) underlying DBMs, we derive a class of models, *inflationary flows,* that uniquely and deterministically map high-dimensional data to a lower-dimensional Gaussian distribution via ODE integration. This map is both invertible and neighborhood-preserving, with controllable numerical error, with the result that uncertainties in the data are correctly propagated to the latent space. We demonstrate how such maps can be learned via standard DBM training using a novel noise schedule and are effective at both preserving and reducing intrinsic data dimensionality. The result is a class of highly expressive generative models, uniquely defined on a low-dimensional latent space, that afford principled Bayesian inference.

## 1   Introduction

In many fields of science, the aim of statistical inference is not only to estimate model parameters of interest from data but to quantify the *uncertainty* in these estimates. In Bayesian inference, for data $\mathbf{x}$ generated from latent parameters $\mathbf{z}$ via a model $p(\mathbf{x}|\mathbf{z})$, this information is encapsulated in the posterior distribution $p(\mathbf{z}|\mathbf{x})$, computation of which requires evaluation of the often intractable normalizing integral $p(\mathbf{x}) = \int p(\mathbf{x}, \mathbf{z}) \, d\mathbf{z}$. Where accurate uncertainty estimation is required, the gold standard remains sampling-based Markov Chain Monte Carlo (MCMC) methods, which are guaranteed (asymptotically) to produce exact samples from the posterior distribution [1]. However, MCMC methods can be computationally costly and do not readily scale either to large or high-dimensional data sets.

38th Conference on Neural Information Processing Systems (NeurIPS 2024).

Alternatively, methods based on variational inference (VI) attempt to approximate posterior distributions by optimization, minimizing some measure of divergence between the true posterior and a parameterized set of distributions $q_\phi(\mathbf{z}|\mathbf{x})$ [2]. For example, methods like the variational autoencoder (VAE) [3, 4] minimize the Kullback-Leibler (KL) divergence between true and approximate posteriors, producing bidirectional mappings between data and latent spaces. In vanilla VAEs, posterior uncertainty estimates are typically overconfident due to minimization of the reverse (mode-seeking) KL divergence [5, 6]. While some lines of work have sought to mitigate this posterior mismatch problem by utilizing different divergences 7–10, VAEs still tend to produce blurry data reconstructions and non-unique latent spaces without additional assumptions [11–13].

By contrast, normalizing flow (NF) models [14, 15] work by applying a series of bijective transformations to a simple base distribution (usually uniform or Gaussian) to deterministically convert samples to a desired target distribution. While NFs have been successfully used for posterior approximation [16–20] and produce higher-quality samples, the requirement that the Jacobian of each transformation be simple to compute often requires a high number of transformations and, traditionally, these transformations do not alter the the dimensionality of their inputs, resulting in latent spaces with thousands of dimensions. More recent lines of work on *injective flow* models 21–25 address this limitation by allowing practitioners to use flows to learn lower dimensional manifolds from data, but most compression-capable flow models still fail to reach high generative performance on key benchmark image datasets (cf. [23]).

More recently, diffusion-based models (DBMs) [26–33] have been shown to achieve state-of-the-art results in several generative tasks, including image, sound, and text-to-image generation. These models work by stipulating a fixed forward noising process (e.g., a forward stochastic differential equation (SDE)), wherein Gaussian noise is incrementally added to samples of the target data distribution until all information in the original data is degraded. To generate samples from the target distribution, one then needs to simulate the reverse de-noising process (reverse SDE [34]) which requires knowledge of the score of the intermediate "noised" transitional densities. Estimation of this score function across multiple noise levels is the key component of DBM model training, typically using a de-noising score matching objective [35, 28, 30]. Yet, despite their excellent performance as *generative* models, DBMs, unlike VAEs or flows, do not readily lend themselves to *inference*. In particular, because DBMs use a *diffusion* process to transform the data distribution, they fail to preserve local structure in the data (**Figure 1**), and uncertainty under this mapping is high at its endpoint because of continuous noise injection and resultant mixing. Moreover, because the final distribution—Gaussian white noise of the same dimension—must have *higher* entropy than the original data, there is no data compression.

Finally, emerging work on *flow matching* models [36–42] has achieved impressive generative performance on several benchmark image datasets. Such models utilize simple *conditional* distribution families to learn a vector field capable of transporting points between two pre-specified densities. These are closely related to the *probability flow ODE (pfODE)* view of DBMs, and, in fact, have been shown to be equivalent to such models for specific choices of "interpolant" functions and conditional distributions. Despite their exceptional generative performance and deterministic nature, existing flow matching approaches do not allow for compression and, therefore, do not allow practitioners to infer a lower dimensional latent space from data.

Thus, despite tremendous improvements in sample quality, modern generative models do not lend themselves to one of the key modeling goals in scientific applications: calibrated Bayesian inference. Note that while many works focus on *predictive* calibration, how well the inferred marginal $p(\mathbf{x})$ matches real data [43–47], our focus here is on *posterior calibration*, how well $q(\mathbf{z}|\mathbf{x})$ matches the true posterior $p(\mathbf{z}|\mathbf{x})$. We address this challenge by demonstrating how a novel DBM variant that we call *inflationary flows* can, in fact, produce calibrated Bayesian inference in this sense.

**Specifically, our contributions are: First,** focusing on the case of *unconditional* generative models, we show how a previously established link between the SDE defining diffusion models and the probability flow ODE (pfODE) that gives rise to the same Fokker-Planck equation [30] can be used to define a *unique, deterministic* map between the original data and an asymptotically Gaussian distribution. This map is bidirectional, preserves local neighborhoods, and has controllable numerical error, making it suitable for rigorous uncertainty quantification. **Second,** we define two classes of flows that correspond to novel noise injection schedules in the forward SDE of the diffusion model. The first of these preserves a measure of dimensionality, the participation ratio (PR) [48], based on

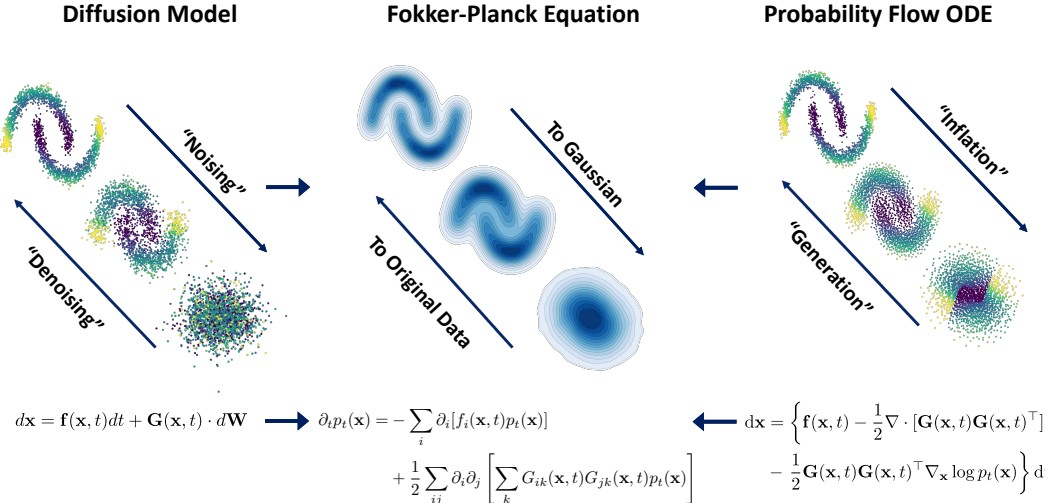

**Diffusion Model**

**Fokker-Planck Equation**

**Probability Flow ODE**

$$dx = f(x, t)dt + G(x, t) \cdot dW \quad\longrightarrow\quad \partial_t p_t(x) = -\sum_i \partial_i [f_i(x, t) p_t(x)]$$

$$+ \frac{1}{2} \sum_{ij} \partial_i \partial_j \left[ \sum_k G_{ik}(x, t) G_{jk}(x, t) p_t(x) \right]$$

$$\longleftarrow\quad dx = \left\{ f(x, t) - \frac{1}{2} \nabla \cdot [G(x, t) G(x, t)^\top] \right.$$

$$\left. - \frac{1}{2} G(x, t) G(x, t)^\top \nabla_x \log p_t(x) \right\} dt$$

Figure 1: **SDE-ODE Duality of diffusion-based models**. The forward (noising) SDE defining the DBM (**left**) gives rise to a sequence of marginal probability densities whose temporal evolution is described by a Fokker-Planck equation (FPE, **middle**). But this correspondence is not unique: the probability flow ODE (pfODE, **right**) gives rise to the *same* FPE. That is, while both the SDE and the pfODE possess the same marginals, the former is noisy and mixing while the latter is deterministic and neighborhood-preserving. Both models require knowledge of the score function $\nabla_x \log p_t(x)$, which can learned by training either model.

second-order data statistics, preventing an effective *increase* in data dimensionality with added noise, while the second flow *reduces* PR, providing *data compression*. We demonstrate experimentally that inflationary flows indeed preserve local neighborhood structure, allowing for sampling-based uncertainty estimation, and that these models continue to provide high-quality generation under compression, even from latent spaces reduced to as little as 0.03% of the nominal data dimensionality. As a result, inflationary flows offer excellent generative performance while affording data compression and accurate uncertainty estimation for scientific applications.

## 2 Three views of diffusion-based models

As with standard DBMs, we assume a data distribution $p_0(x) = p_{\text{data}}(x)$ at time $t = 0$, transformed via a forward noising process defined by the stochastic differential equation [e.g., 26, 28]:

$$dx = f(x, t)dt + G(x, t) \cdot dW, \tag{1}$$

with most DBMs assuming linear drift ($f = f(t)x$) and isotropic noise ($G = \sigma(t)\mathbb{1}$) that monotonically increases over time [49]. As a result, for $\int_0^T \sigma(T)dt \gg \sigma_{data}$, $p_T(x)$ becomes essentially indistinguishable from an isotropic Gaussian (**Figure 1, left**). DBMs work by learning an approximation to the reverse SDE [34, 28–30, 50],

$$dx = \{f(x, t) - \nabla \cdot [G(x, t)G(x, t)^\top] - G(x, t)G(x, t)^\top \nabla_x \log p_t(x)\}dt + G(x, t) \cdot d\bar{W}, \tag{2}$$

where $\bar{W}$ is time-reversed Brownian motion. In practice, this requires approximating the score function $\nabla_x \log p_t(x)$ by incrementally adding noise according to the schedule $\sigma(t)$ of the forward process and then requiring that denoising by (2) match the original sample. The fully trained model then generates samples from the target distribution by starting with $x_T \sim \mathcal{N}(0, \sigma^2(T)\mathbb{1})$ and integrating (2) in reversed time.

As previously shown, this diffusive process gives rise to a series of marginal distributions $p_t(x)$ satisfying a Fokker-Planck equation (**Figure 1, middle**) [30, 49],

$$\partial_t p_t(x) = -\sum_i \partial_i [f_i(x, t) p_t(x)] + \frac{1}{2} \sum_{ij} \partial_i \partial_j \left[ \sum_k G_{ik}(x, t) G_{jk}(x, t) p_t(x) \right], \tag{3}$$

where $\partial_i \equiv \frac{\partial}{\partial x_i}$. In the "variance preserving" noise schedule of [30], (3) has as its stationary solution an isotropic Gaussian distribution. This "distributional" perspective views the forward process as a means of transforming the data into an easy-to-sample form (as with normalizing flows) and the reverse process as a means of data generation.

However, in addition to the SDE and FPE perspectives, Song et al. [30] also showed that (3) is satisfied by the marginals of a different process with no noise term, the so-called *probability flow ODE* (pfODE):

$$\mathrm{d}\mathbf{x} = \left\{ \mathbf{f}(\mathbf{x}, t) - \frac{1}{2}\nabla \cdot [\mathbf{G}(\mathbf{x}, t)\mathbf{G}(\mathbf{x}, t)^\top] - \frac{1}{2}\mathbf{G}(\mathbf{x}, t)\mathbf{G}(\mathbf{x}, t)^\top \nabla_\mathbf{x} \log p_t(\mathbf{x}) \right\} \mathrm{d}t. \qquad (4)$$

Unlike (1), this process is deterministic, and data points evolve smoothly (**Figure 1, right**), resulting in a flow that preserves local neighborhoods. Moreover, the pfODE is uniquely defined by $\mathbf{f}(\mathbf{x}, t)$, $\mathbf{G}(\mathbf{x}, t)$, and the score function. This connection between the marginals satisfying the SDEs of diffusion processes and *deterministic flows* described by an equivalent ODE has also been recently explored in the context of flow matching models [36–42], a connection on which we elaborate in **Section 7**.

In the following sections, we show how this pfODE, constructed using a score function estimated by training the corresponding DBM, can be used to map points from $p_{\mathrm{data}}(\mathbf{x})$ to a compressed latent space in a manner that affords accurate uncertainty quantification.

## 3   Inflationary flows

As argued above, the probability flow ODE offers a means of deterministically transforming an arbitrary data distribution into a simpler form via a score function learnable through DBM training. Here, we introduce a specialized class of pfODEs, *inflationary flows*, that follow from an intuitive picture of local dynamics and asymptotically give rise to stationary Gaussian solutions of (3).

We begin by considering a sequence of marginal transformations in which points in the original data distribution are convolved with Gaussians of increasingly larger covariance $\mathbf{C}(t)$:

$$p_t(\mathbf{x}) = p_0(\mathbf{x}) * \mathcal{N}(\mathbf{x}; \mathbf{0}, \mathbf{C}(t)). \qquad (5)$$

It is straightforward to show (**Appendix A.1**) that this class of time-varying densities satisfies (3) when $\mathbf{f} = \mathbf{0}$ and $\mathbf{GG}^\top = \dot{\mathbf{C}}$. This can be viewed as a process of deterministically "inflating" each point in the data set, or equivalently as smoothing the underlying data distribution on ever coarser scales, similar to denoising approaches to DBMs [51, 52]. Eventually, if the smoothing kernel grows much larger than $\mathbf{\Sigma}_0$, the covariance in the original data, total covariance $\mathbf{\Sigma}(t) \equiv \mathbf{\Sigma}_0 + \mathbf{C}(t) \to \mathbf{C}(t)$, $p_t(\mathbf{x}) \approx \mathcal{N}(\mathbf{0}, \mathbf{C}(t))$, and all information has been removed from the original distribution. However, because it is numerically inconvenient for the variance of the asymptotic distribution $p_\infty(\mathbf{x})$ to grow much larger than that of the data, we follow previous work in adding a time-dependent coordinate rescaling $\tilde{\mathbf{x}}(t) = \mathbf{A}(t) \cdot \mathbf{x}(t)$ [30, 49], which results in an asymptotic solution $p_\infty(\mathbf{x}) = \mathcal{N}(\mathbf{0}, \mathbf{A}\mathbf{\Sigma}\mathbf{A}^\top)$ of the corresponding Fokker-Planck equation when $\dot{\mathbf{\Sigma}} = \dot{\mathbf{C}}$ and $\dot{\mathbf{A}}\mathbf{\Sigma}\mathbf{A}^\top + \mathbf{A}\mathbf{\Sigma}\dot{\mathbf{A}}^\top = \mathbf{0}$ (**Appendix A.2**). Together, these assumptions give rise to the pfODE (**Appendix A.3**):

$$\frac{\mathrm{d}\tilde{\mathbf{x}}}{\mathrm{d}t} = \mathbf{A}(t) \cdot \left( -\frac{1}{2}\dot{\mathbf{C}}(t) \cdot \nabla_\mathbf{x} \log p_t(\mathbf{x}) \right) + \left( \dot{\mathbf{A}}(t) \cdot \mathbf{A}^{-1}(t) \right) \cdot \tilde{\mathbf{x}}, \qquad (6)$$

where the score function is evaluated at $\mathbf{x} = \mathbf{A}^{-1} \cdot \tilde{\mathbf{x}}$. Notably, (6) is equivalent to the general pfODE form given in [49] in the case both $\mathbf{C}(t)$ and $\mathbf{A}(t)$ are isotropic (**Appendix A.4**), with $\mathbf{C}(t)$ playing the role of injected noise and $\mathbf{A}(t)$ the role of the scale schedule. In the following sections, we will show how to choose both of these in ways that either preserve or reduce intrinsic data dimensionality.

### 3.1   Dimension-preserving flows

In standard DBMs, the final form of the distribution $p_T(\mathbf{x})$ approximates an isotropic Gaussian distribution, typically with unit variance. As a result, these models *increase* the effective dimensionality of the data, which may begin as a low-dimensional manifold embedded within $\mathbb{R}^d$. Thus, even

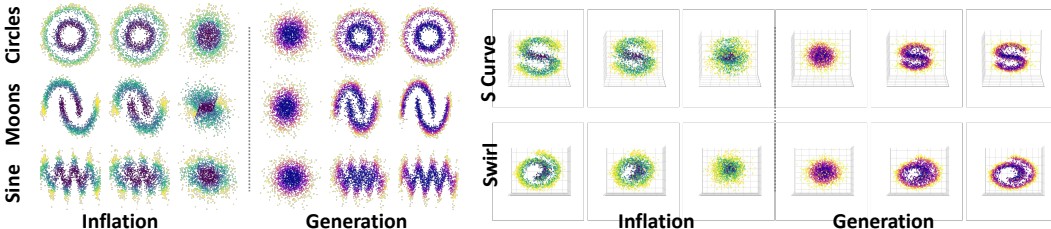

**Dimension Preserving Toy Simulations**

Figure 2: **Dimension-preserving flows for toy datasets.** Numerical simulations of dimension-preserving flows for five sample toy datasets. Left sequences of sub-panels show results for integrating the pfODE forward in time (inflation); right sub-panels show results of integrating the same system backwards in time (generation) (**Appendix B.3**). Simulations were conducted with score approximations obtained from neural networks trained on each respective toy dataset (**Appendix B.4.1**).

maintaining intrinsic data dimensionality requires both a definition of dimensionality and a choice of flow that preserves this dimension. In this work, we consider a particularly simple measure of dimensionality, the participation ratio (PR), first introduced by Gao et al. [48]:

$$\text{PR}(\mathbf{\Sigma}) = \frac{\text{tr}(\mathbf{\Sigma})^2}{\text{tr}(\mathbf{\Sigma}^2)} = \frac{(\sum_i \sigma_i^2)^2}{\sum_i \sigma_i^4} \tag{7}$$

where $\mathbf{\Sigma}$ is the covariance of the data with eigenvalues $\{\sigma_i^2\}$. PR is invariant to linear transforms of the data, depends only on second-order statistics, is 1 when $\mathbf{\Sigma}$ is rank-1, and is equal to the nominal dimensionality $d$ when $\mathbf{\Sigma} \propto \mathbb{1}_{d \times d}$. In **Appendix C.1** we report this value for several benchmark image datasets, confirming that in all cases, PR is substantially lower than the nominal data dimensionality.

To construct flows that preserve this measure of dimension, following (5), we write total variance as $\mathbf{\Sigma}(t) = \text{diag}(\boldsymbol{\sigma}^2(t)) = \mathbf{C}(t) + \mathbf{\Sigma}_0$, where $\mathbf{\Sigma}_0$ is the original data covariance and $\mathbf{C}(t)$ is our time-dependent smoothing kernel. Moreover, we will choose $\mathbf{C}(t)$ to be diagonal in the eigenbasis of $\mathbf{\Sigma}_0$ and work in that basis, in which case $\mathbf{\Sigma}(t) = \text{diag}(\boldsymbol{\sigma^2}(t))$ and we have (**Appendix A.6**):

$$d\text{PR} = 0 \iff \left(\mathbf{1} - \text{PR}(\boldsymbol{\sigma^2})\frac{\boldsymbol{\sigma^2}}{\sum_k \sigma_k^2}\right) \cdot d\boldsymbol{\sigma^2} = 0. \tag{8}$$

The simplest solution to this constraint is a proportional inflation, $\frac{d}{dt}(\boldsymbol{\sigma^2}) = \rho \boldsymbol{\sigma^2}$, along with a rescaling along each principal axis:

$$C_{jj}(t) = \sigma_j^2(t) - \sigma_{0j}^2 = \sigma_{0j}^2(e^{\rho t} - 1) \qquad A_{jj}(t) = \frac{A_{0j}}{\sigma_j(t)} = \frac{A_{0j}}{\sigma_{0j}}e^{-\rho t/2}. \tag{9}$$

As with other flow models based on physical processes like diffusion [26] or electrostatics [53, 54], our use of the term *inflationary flows* for these choices is inspired by cosmology, where a similar process of rapid expansion exponentially suppresses local fluctuations in background radiation density [55]. However, as a result of our coordinate rescaling, the effective covariance $\tilde{\mathbf{\Sigma}} = \mathbf{A}\mathbf{\Sigma}\mathbf{A}^\top = \text{diag}(A_{0j}^2)$ remains constant (so $d\tilde{\boldsymbol{\sigma}}^2 = \mathbf{0}$ trivially), and the additional conditions of **Appendix A.2** are satisfied, such that $\mathcal{N}(\mathbf{0}, \tilde{\mathbf{\Sigma}})$ is a stationary solution of the relevant rescaled Fokker-Planck equation. As **Figure 2** shows, these choices result in a version of (6) that smoothly maps nonlinear manifolds to Gaussians and can be integrated in reverse to generate samples of the original data.

## 3.2 Dimension-reducing flows

In the previous section, we saw that isotropic inflation preserves intrinsic data dimensionality as measured by PR. Here, we generalize and consider *anisotropic* inflation at different rates along each of the eigenvectors of $\mathbf{\Sigma}$: $\frac{d}{dt}(\boldsymbol{\sigma^2}) = \rho \mathbf{g} \odot \boldsymbol{\sigma^2}$. In addition, we denote $g_* \equiv \max(\mathbf{g})$, so that the

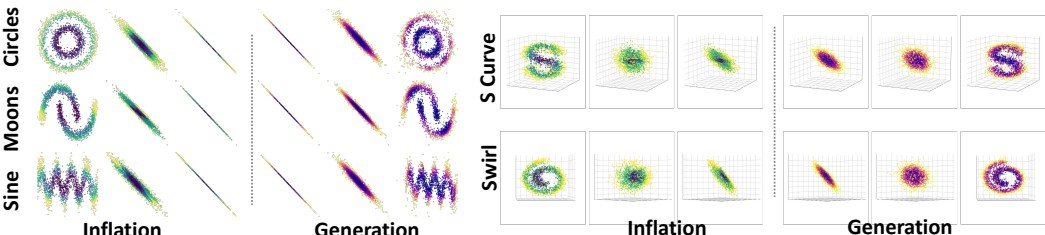

**Dimension Reducing Toy Simulations**

Figure 3: **Dimension-reducing flows for toy datasets.** Numerical simulations of dimension-reducing flows for the same five datasets as in **Figure 2**. For 2D datasets, we showcase reduction from two to one dimension, while 3D datasets are reduced to two dimensions. Colors and layouts are the same as in **Figure 2**, with scores again estimated using neural networks trained on each example. Additional results showcasing (1) similar flows further compressing two-dimensional manifolds embedded in $D = 3$ space, and (2) effects of adopting different scaling schemes for target data are given in **Appendices C.2.2** and **C.2.3**, respectively.

fastest inflation rate is $\rho g_*$. Then, if we take $g_i = g_*$ for $i \in \{i_1, i_2, \ldots i_K\}$ and $g_i < g_*$ for the other dimensions,

$$\mathrm{PR}(\boldsymbol{\Sigma}(t)) = \frac{(\sum_i \sigma_{0i}^2 e^{(g_i - g_*)\rho t})^2}{\sum_i (\sigma_{0i}^2 e^{(g_i - g_*)\rho t})^2} \xrightarrow{t \to \infty} \frac{(\sum_{k=1}^K \sigma_{0i_k}^2)^2}{\sum_{j=1}^K \sigma_{0i_j}^4} \tag{10}$$

which is the dimension that would be achieved by simply truncating the original covariance matrix in a manner set by our choice of $\mathbf{g}$. Here, unlike in (9), we do not aim for rescaling to compensate for expansion along each dimension, since that would undo the effect of differential inflation rates. Instead, we choose a single global rescaling factor $\alpha(t) \propto A_0 \exp(-\rho g_* t/2)$, leading to a Gaussian asymptotic solution with the original data covariance in dimensions $i \in \{i_1, i_2, \ldots i_K\}$.

Two additional features of this class of flows are worth noting: First, the final scale ratio of preserved to shrunken dimensions for finite integration times $T$ is governed by the quantity $e^{\rho(g_* - g_i)T}$ in (10). For good compression, we want this number to be very large, but as we show in **Appendix A.4**, this corresponds to a maximum injected noise of order $e^{\rho(g_* - g_i)T/2}$ in the equivalent DBM. That is, the compression one can achieve with inflationary flows is constrained by the range of noise levels over which the score function can be accurately estimated, and this is quite limited in typical models. Second, despite the appearance given by (10), the corresponding flow *is not* simply a linear projection to the top $K$ principal components: though higher PCs are selectively removed by dimension-reducing flows via exponential shrinkage, individual particles are repelled by *local* density as captured by the score function (6), and this term couples different dimensions even when $\mathbf{C}$ and $\mathbf{A}$ are diagonal. Thus, the final positions of particles in the retained dimensions depend on their initial positions in the full space, producing a nonlinear map (**Figure 3**).

## 4 Score function approximation from DBMs

Having chosen inflation and rescaling schedules, the last component needed for the pfODE (6) is the score function $\mathbf{s}(\mathbf{x}, t) \equiv \nabla_{\mathbf{x}} \log p_t(\mathbf{x})$. Our strategy will be to exploit the correspondence described above between diffusion models (1) and pfODEs (4) that give rise to the same marginals (3). That is, we will learn an approximation to $\mathbf{s}(\mathbf{x}, t)$ by fitting the DBM corresponding to our desired pfODE, since both make use of the same score function.

Briefly, in line with previous work on DBMs [49], we train neural networks to estimate a de-noised version, $\mathbf{D}(\mathbf{x}, \mathbf{C}(t))$, of a noise-corrupted data sample $\mathbf{x}$ given noise level $\mathbf{C}(t)$ (cf. **Appendix A.4** for the correspondence between $\mathbf{C}(t)$ and noise). That is, we model $\mathbf{D}_\theta(\mathbf{x}, \mathbf{C}(t))$ using a neural network and train it by minimizing a standard $L_2$ de-noising error:

$$\mathbb{E}_{\mathbf{y} \sim \text{data}} \mathbb{E}_{\mathbf{n} \sim \mathcal{N}(\mathbf{0}, \mathbf{C}(t))} \|\mathbf{D}(\mathbf{y} + \mathbf{n}; \mathbf{C}(t)) - \mathbf{y}\|_2^2 \tag{11}$$

De-noised outputs can then be used to compute the desired score term using $\nabla_{\mathbf{x}} \log p(\mathbf{x}, \mathbf{C}(t)) = \mathbf{C}^{-1}(t) \cdot (\mathbf{D}(\mathbf{x}; \mathbf{C}(t)) - \mathbf{x})$ [30, 49]. Moreover, as in [49], we also adopt a series of preconditioning

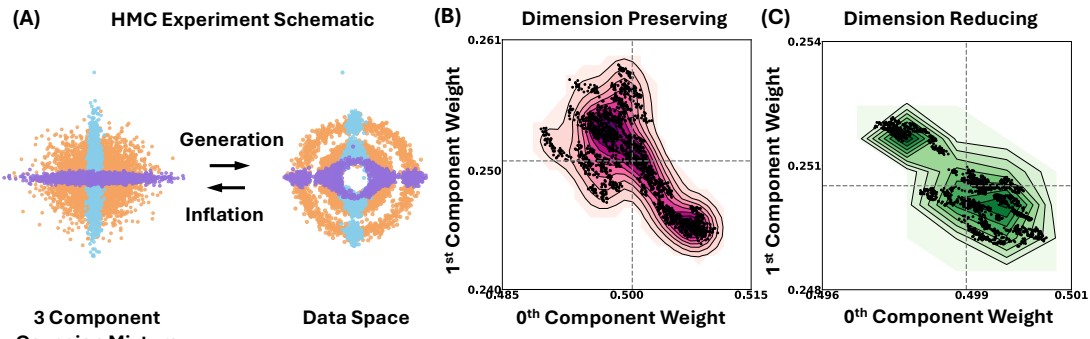

Figure 4: **Calibration experiments.** To assess error in our posterior model estimates, we used Hamiltonian Monte Carlo (HMC) to perform inference in one of our toy datasets (2D circles). Drawing samples from a 3-component Gaussian Mixture Model (GMM) prior, we integrated the generative process backward in time to obtain corresponding data space samples (**A**, components shown in orange, blue, and purple). We then used HMC to obtain posterior samples from the posterior distribution over the weights of the GMM components. (**B, C**) Kernel density estimates from the joint posterior samples over the mixture distribution weights in the dimension-preserving and dimension-reducing cases. Dashed vertical and horizontal lines indicate posterior means for each component. Reference ground-truth weights were $\mathbf{w} = [0.5, 0.25, 0.25]$.

factors aimed at making training with the above $L_2$ loss and our noising scheme more amenable to gradient descent techniques (**Appendix B.1**).

## 5  Calibrated uncertainty estimates from inflationary flows

Several previous lines of work [43–47] have focused on assessing how well model-predicted marginals $p(\mathbf{x})$ match real data (i.e., the *predictive* calibration case). Though we do compare our models' predictive calibration performance against existing injective flow models (**Table 3**), here we are primarily focused on quantifying error in unconditional posterior inference. That is, we are interested in quantifying the mismatch between inferred posteriors $q(\mathbf{z}|\mathbf{x})$ and true posteriors $p(\mathbf{z}|\mathbf{x})$, especially in contexts where the true generative model is unknown and must be learned from data. This is by far the most common scenario in modern generative models like VAEs, flows, and GANs.

As with other implicit models, our inflationary flows provide a deterministic link between complex data and simplified distributions with tractable sampling properties. This mapping requires integrating the pfODE (6) for a given choice of $\mathbf{C}(t)$ and $\mathbf{A}(t)$ and an estimate of the score function of the original data. As a result, sampling-based estimates of uncertainty are trivial to compute: given a prior $\pi(\mathbf{x})$ over the data (e.g., a Gaussian ball centered on a particular example $\mathbf{x}_0$), this can be transformed into an uncertainty on the dimension-reduced space by sampling $\{\mathbf{x}_j\} \sim \pi(\mathbf{x})$ and integrating (6) forward to generate samples from $\int p(\mathbf{x}_T|\mathbf{x}_0)\pi(\mathbf{x}_0)\,\mathrm{d}\mathbf{x}_0$. As with MCMC, these samples can be used to construct either estimates of the posterior or credible intervals. Moreover, because the pfODE is unique given $\mathbf{C}$, $\mathbf{A}$, and the score, the model is *identifiable* when conditioned on these choices.

The only potential source of error, apart from Monte Carlo error, in the above procedure arises from the fact that the score function used in (6) is only an *estimate* of the true score. To assess whether integrating noisy estimates of the score could produce errant posterior samples, we conducted the experiment showcased in **Figure 4A** (**Appendix B.6**). Briefly, we constructed a Gaussian Mixture Model (GMM) prior with three pre-specified components (**Appendix B.6**) from which we drew samples of $\mathbf{z}$, integrating backwards in time using our trained pfODE networks to construct corresponding observed data points $\mathbf{x}$. We then utilized Hamiltonian Monte Carlo (HMC) 1, 56–58 to obtain posterior samples for the GMM component weights. As shown in **Figure 4B, C**, the resulting posterior correctly covers the original ground-truth values, suggesting that numerical errors in score estimates, at least in this simplified scenario, do not appreciably accumulate. This is likely because, empirically, score estimates do not appear to be strongly auto-correlated in time (**Appendix C.3**),

suggesting that $\hat{\mathbf{s}}(\mathbf{x}, t)$ is well approximated as a scaled colored noise process and the corresponding pfODE as an SDE. In such a case, standard theorems for SDE integration show that while errors due to noise do accumulate, these can be mitigated by a careful choice of integrator and ultimately controlled by reducing step size [59, 60]. In addition, we verified this empirically in both low-dimensional examples (**Figure 4**, **Appendices B.6, C.2.1**) and with round-trip integration of the pfODE in high-dimensional datasets (**Tables 1, 2**, **Appendix B.5.1**).

## 6 Experiments

For the PR-Reducing flows, the final scale ratio between preserved vs. shrunken dimensions for finite integration times is dependent on the quantity $e^{\rho(g_* - g_i)T}$. Therefore, for fixed end integration time $T$ and rate $\rho$, this scaling is dictated by $g_* - g_i$, which we call the "inflation gap" (IG), **Appendix B.2**. As this inflation gap increases, compressed dimensions are shrunken to a greater extent, and the denoising networks are required to amortize score estimation over wider noise scales, a harder learning problem. Therefore, for our proposed model, compression should be understood *both* in terms of the number of dimensions being preserved and the size of this inflation gap.

To assess how these two factors affect model performance, we performed two sets of experiments on two benchmark image datasets (CIFAR-10 [61] and AFHQv2 [62]; **Appendix B.4.2**; code: [63]; project website: [64]). In the first set of experiments, we fixed $T$, $\rho$, and the inflation gap (IG $= 1.02$) while varying only the number of preserved dimensions $d$ between $d = 1$ (compression to $\approx 0.03\%$) and $d = 3072$ (no compression) for both datasets. For the second set of experiments, we worked with the AFHQv2 dataset and fixed $T$, $\rho$, and $d = 2$, while varying the inflation gap (IG $= 1.10, 1.25, 1.35, 1.50$). In **Tables 1** and **2** we showcase Frechet Inception Distance (FID) scores [65] (mean $\pm 2\sigma$ over 3 independently generated sets of images, each with 50,000 samples) and round-trip integration mean squared errors (mean MSE $\pm 2\sigma$ over 3 randomly sampled sets of images, each with 10,000 samples) for each ($d$, IG) combination explored (**Appendices B.5.1, B.5.2, B.5.4**). **Figures 5**, **6**, and **7** showcase 24 randomly generated images (top rows) along with round-trip integration results for 8 randomly sampled images (bottom rows), across select ($d$, IG) combinations.

Finally, we also compared our *inflationary flows* (IFs) model generative performance on CIFAR-10 against three existing *injective flow* model baselines (**Appendix B.5.2**) — M-Flows [21], Rectangular Flows (RFs) [22], and Canonical Manifold Flows (CMF) [23] — for different numbers of preserved dimensions ($d = 30, 40, 62$). **Table 3** showcases best FID scores (out of 3 independently generated sets of images, each with 10,000 samples) for each such experiment. For these comparison experiments, we fixed IG=1.02 when training our networks for the different $d$ values.

As a general trend, increasing the number of preserved dimensions at a constant inflation gap led to improvements in generative quality (lower FID scores) and reduced MSE (**Table 1**). However, some schedules we assessed are not entirely consistent with this trend. We hypothesize this is at least partially due to variance arising from different network initializations for each schedule (**Appendix B.5.3**), as well as differences between the two datasets explored here. As expected, increasing inflation gap while maintaining the number of preserved dimensions leads to worsened generative performance (higher FID scores, **Table 2**). Finally, in terms of predictive calibration, our model provides substantial gains when compared to existing *injective flow* model baselines (**Table 3**).

Table 1: FID and round-trip MSE (mean $\pm 2\sigma$) at 1.02 Inflation Gap (IG)

| | AFHQv2 | | | CIFAR-10 | |
| Dimensions | FID | MSE | Dimensions | FID | MSE |
| --- | --- | --- | --- | --- | --- |
| 1 | $12.65 \pm 0.07$ | $1.47 \pm 0.07$ | 1 | $20.76 \pm 0.09$ | $1.07 \pm 0.10$ |
| 2 | $11.95 \pm 0.06$ | $1.55 \pm 0.21$ | 2 | $21.29 \pm 0.04$ | $0.81 \pm 0.11$ |
| 30 | $13.64 \pm 0.02$ | $3.79 \pm 0.13$ | 30 | $23.36 \pm 0.14$ | $2.21 \pm 0.08$ |
| 62 | $14.05 \pm 0.18$ | $5.32 \pm 0.18$ | 62 | $23.30 \pm 0.19$ | $2.27 \pm 0.24$ |
| 307 | $15.64 \pm 0.10$ | $3.33 \pm 0.13$ | 307 | $28.07 \pm 0.13$ | $0.71 \pm 0.02$ |
| 615 | $14.63 \pm 0.07$ | $2.42 \pm 0.18$ | 615 | $24.49 \pm 0.27$ | $0.29 \pm 0.03$ |
| 1536 | $13.36 \pm 0.12$ | $0.14 \pm 0.03$ | 1536 | $17.44 \pm 0.16$ | $0.16 \pm 0.06$ |
| 3041 | $13.97 \pm 0.13$ | $0.28 \pm 0.06$ | 3041 | $16.60 \pm 0.05$ | $0.30 \pm 0.02$ |
| 3072 | $11.90 \pm 0.08$ | $0.38 \pm 0.04$ | 3072 | $17.01 \pm 0.10$ | $0.22 \pm 0.03$ |

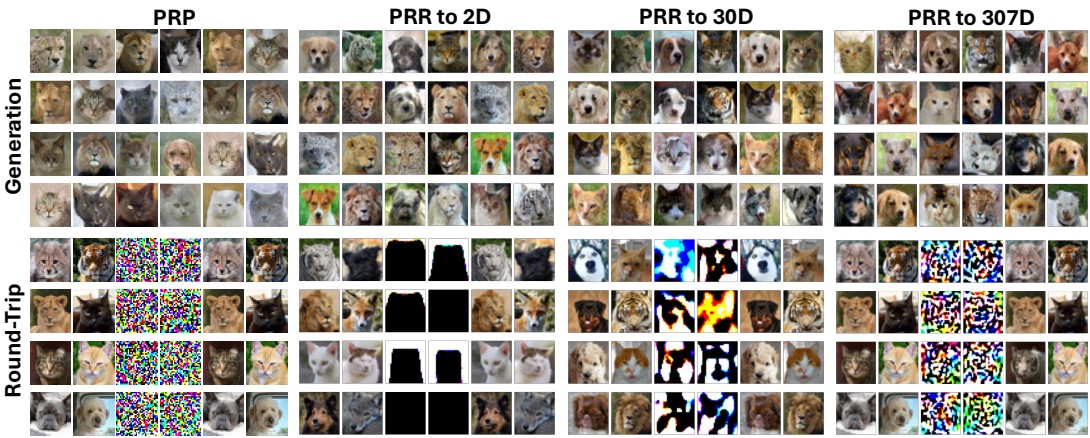

Figure 5: **Generation and round-trip experiments for AFHQv2 at IG=1.02 and varying number of preserved dimensions**. **Top row:** Generated samples for select flow schedules (PR-Preserving (PRP), PR-Reducing to 2D ($\approx 0.07\%$), 30D($\approx 1\%$), and 307D($\approx 10\%$), at 1.02 IG. **Bottom row:** Results for round-trip experiments under same schedules. Leftmost columns are original samples, middle columns are samples mapped to Gaussian latent spaces, and rightmost columns are recovered samples.

Table 2: FID and round-trip MSE (mean $\pm 2\sigma$) for AFHQv2 at varying Inflation Gaps (IG)

| Dimensions | IG | FID | MSE |
|---|---|---|---|
| 2 | 1.02 | $11.95 \pm 0.06$ | $1.55 \pm 0.21$ |
| 2 | 1.10 | $13.98 \pm 0.13$ | $1.35 \pm 0.08$ |
| 2 | 1.25 | $17.84 \pm 0.15$ | $1.65 \pm 0.09$ |
| 2 | 1.35 | $34.68 \pm 0.37$ | $1.19 \pm 0.18$ |
| 2 | 1.50 | $107.64 \pm 0.43$ | $0.11 \pm 0.02$ |

## 7  Discussion

Here, we have proposed a new type of implicit probabilistic model based on the probability flow ODE (pfODE) in which it is possible to perform calibrated, identifiable Bayesian inference on a reduced-dimension latent space via sampling and integration. To do so, we have leveraged a correspondence between pfODEs and diffusion-based models by means of their associated Fokker-Planck equations, and we have demonstrated that such models continue to produce high-quality generated samples even when latent spaces are as little as 0.03% of the nominal data dimension. More importantly, the uniqueness and controllable error of the generative process make these models an attractive approach in cases where accurate uncertainty estimates are required.

**Limitations:** One limitation of our model is its reliance on the participation ratio (7) as a measure of dimensionality. Because PR relies only on second-order statistics and our proposals (9) are formulated in the data eigenbasis, our method tends to favor the top principal components of the data when reducing dimension. However, as noted above, this is not simply a truncation to the lowest principal components, since dimensions still mix via coupling to the score function in (6). Nonetheless, solutions to the condition (8) that preserve (or reduce) more complex dimensionality measures might lead to even stronger compressions for curved manifolds (**Appendix C.2.2**), and more sophisticated choices for noise and rescaling schedules in (6) might lead to compressions that do not simply remove information along fixed axes, more similar to [66]. That is, we believe much more interesting classes of flows are possible. A second limitation is that mentioned in **Section 3.2** and in our experiments: our schedule requires training DBMs over much larger ranges of noise than are typically used, and this results in noticeable tradeoffs in compression performance as the inflation gap and number of preserved dimensions are varied.

Table 3: FID score comparison with injective flows for CIFAR-10

| Dimensions Preserved | IFs (IG=1.02) | M-Flow | RFs | CMFs |
|---:|:---:|:---:|:---:|:---:|
| 30 | **23.3** | 541.2 | 544.0 | 532.6 |
| 40 | **24.3** | 535.7 | 481.3 | 444.6 |
| 62 | **23.2** | 280.9 | 280.8 | 287.9 |

**Related work:** This work draws on several related lines of research, including work on using DBMs as likelihood estimation machines [50, 67, 31], relations with normalizing flows and hierarchical VAEs [67, 33, 68], *injective flow* models [21–25], and generative flow networks [69]. By contrast, our focus is on the use of DBMs to learn score functions estimates for implicit probabilistic models, with the ultimate goal of performing accurate posterior inference. In this way, it is also closely related to work on denoising models [51, 52, 66, 70] that cast that process in terms of statistical inference and to models that use DBMs for de-blurring and in-painting [71, 72]. However, this work is distinct from several models that use reversal of deterministic transforms to train generative models [73–76]. Whereas those models work by removing information from each sample $\mathbf{x}$, our proposal relies critically on adjusting the local density of samples with respect to one another, moving the marginal distribution toward a Gaussian.

Our work is also similar to methods that use DBMs to construct samplers for unnormalized distributions [77–81]. Whereas we begin with samples from the target distribution and aim to learn latent representations, those studies start with a pre-specified form for the target distribution and aim to generate samples. Other groups have also leveraged sequential Monte Carlo (SMC) techniques to construct new types of denoising diffusion samplers for, e.g., conditional generation [82–84]. While our goals are distinct, we believe that the highly simplified Gaussian distribution of our latent spaces may potentially render joint and conditional generation more tractable in future models. Finally, while many prior studies have considered compressed representations for diffusion models [85–88], typically in an encoder-decoder framework, the focus there has been on generative quality, not inference. Along these lines, the most closely related to our work here is [89], which considered diffusion along linear subspaces as a means of improving sample quality in DBMs, though there again, the focus was on improving generation and computational efficiency, not statistical inference.

Yet another line of work closely related to ours is the emerging literature on *flow matching* [36–38, 90] models, which utilize a simple, time-differentiable, "interpolant" function to specify *conditional* families of distributions that continuously map between specified initial and final densities. That is, the interpolant functions define flows that map samples from a base distribution $\rho_0(\mathbf{x})$ to samples from a target distribution $\rho_1(\mathbf{x})$. Typically, these approaches rely on a simple quadratic objective that attempts to match the *conditional* flow field, which can be computed in closed form without needing to integrate the corresponding ODE. As shown in **Appendix A.5**, the pfODEs obtained using our proposed scaling and noising schedules are *equivalent* to the ODEs obtained by using the "Gaussian paths formulation" from [36] when the latter are generalized to full covariance matrices. As a result, our models are amenable to training using flow-matching techniques, suggesting that faster training and inference schemes may be possible through leveraging connections between flow matching and optimal transport [40, 42, 41, 38]

**Broader impacts:** Works like this one that focus on improving generative models risk contributing to an increasingly dangerous set of tools capable of creating misleading, exploitative, or plagiarized content. While this work does not seek to improve the quality of data generation, it does propose a set of models that feature more informative latent representations of data, which could potentially be leveraged to those ends. However, this latent data organization may also help to mitigate certain types of content generation by selectively removing, prohibiting, or flagging regions of the compressed space corresponding to harmful or dangerous content. We believe this is a promising line of research that, if developed further, might help address privacy and security concerns raised by generative models.

## Acknowledgments and Disclosure of Funding

This work was supported by NIH grants F30MH129086 (DdA) and 1RF1DA056376 (JMP).

We also thank Eero Simoncelli for comments and discussion on an early version of this work.

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

# A   Appendix: Additional Details on Model and Preliminaries

## A.1   Derivation of the inflationary Fokker-Planck Equation

We start with derivatives of the smoothing kernel $\kappa(\mathbf{x}, t) \equiv \mathcal{N}(\mathbf{x}; \boldsymbol{\mu}, \mathbf{C}(t))$:

$$\partial_t \kappa(\mathbf{x}, t) = \left[ -\frac{1}{2} \mathrm{tr}(\mathbf{C}^{-1}\dot{\mathbf{C}}) + \frac{1}{2} \mathrm{tr}\left( \mathbf{C}^{-1}(\mathbf{x} - \boldsymbol{\mu})(\mathbf{x} - \boldsymbol{\mu})^\top \mathbf{C}^{-1}\dot{\mathbf{C}} \right) \right] \kappa(\mathbf{x}, t) \tag{12}$$

$$\nabla \kappa = -\mathbf{C}^{-1}(\mathbf{x} - \boldsymbol{\mu})\kappa \tag{13}$$

$$\partial_i \partial_j \kappa = \left[ [\mathbf{C}^{-1}(\mathbf{x} - \boldsymbol{\mu})]_i [\mathbf{C}^{-1}(\mathbf{x} - \boldsymbol{\mu})]_j - (\mathbf{C}^{-1})_{ij} \right] \kappa \tag{14}$$

and combine this with (5) to calculate terms in (3):

$$\partial_t p = p_0(\mathbf{x}) * \partial_t \kappa(\mathbf{x}, \mathbf{t}) \tag{15}$$

$$= p_0 * \left[ -\frac{1}{2} \mathrm{tr}(\mathbf{C}^{-1}\dot{\mathbf{C}}) + \frac{1}{2} \mathrm{tr}\left( \mathbf{C}^{-1}(\mathbf{x} - \boldsymbol{\mu})(\mathbf{x} - \boldsymbol{\mu})^\top \mathbf{C}^{-1}\dot{\mathbf{C}} \right) \right] \kappa \tag{16}$$

$$-\sum_i \partial_i [f_i p] = -p_0 * \sum_i [(\partial_i f_i)\kappa - f_i (\mathbf{C}^{-1}(\mathbf{x} - \boldsymbol{\mu}))_i \kappa] \tag{17}$$

$$\frac{1}{2} \sum_{ij} \partial_i \partial_j \left[ \sum_k G_{ik} G_{jk} p \right] = \frac{1}{2} p_0 * \sum_{ij} \left[ \partial_i \partial_j \left[ \sum_k G_{ik} G_{jk} \right] \kappa \right. \tag{18}$$

$$- 2 \partial_j \left[ \sum_k G_{ik} G_{jk} \right] (\mathbf{C}^{-1}(\mathbf{x} - \boldsymbol{\mu}))_i \kappa$$

$$\left. + \left[ \sum_k G_{ik} G_{jk} \right] \left[ [\mathbf{C}^{-1}(\mathbf{x} - \boldsymbol{\mu})]_i [\mathbf{C}^{-1}(\mathbf{x} - \boldsymbol{\mu})]_j - (\mathbf{C}^{-1})_{ij} \right] \kappa \right].$$

Assuming $\mathbf{f} = \mathbf{0}$ and $\partial_i G_{jk}(\mathbf{x}, t) = 0$ then gives the condition

$$-\frac{1}{2} \mathrm{tr}(\mathbf{C}^{-1}\dot{\mathbf{C}}) + \frac{1}{2} \mathrm{tr}\left( \mathbf{C}^{-1}(\mathbf{x} - \boldsymbol{\mu})(\mathbf{x} - \boldsymbol{\mu})^\top \mathbf{C}^{-1}\dot{\mathbf{C}} \right) =$$

$$-\frac{1}{2} \mathrm{tr}(\mathbf{C}^{-1}\mathbf{G}\mathbf{G}^\top) + \frac{1}{2} \mathrm{tr}\left( \mathbf{C}^{-1}(\mathbf{x} - \boldsymbol{\mu})(\mathbf{x} - \boldsymbol{\mu})^\top \mathbf{C}^{-1}\mathbf{G}\mathbf{G}^\top \right) \tag{19}$$

which is satisfied when $\mathbf{G}\mathbf{G}^\top(\mathbf{x}, t) = \dot{\mathbf{C}}(t)$.

## A.2   Stationary solutions of the inflationary Fokker-Planck Equation

Starting from the unscaled Fokker-Planck Equation corresponding to the process of **Appendix A.1**

$$\partial_t p_t(\mathbf{x}) = \frac{1}{2} \sum_{ij} \dot{C}_{ij}(t) \partial_i \partial_j p_t(\mathbf{x}), \tag{20}$$

we introduce new coordinates $\tilde{\mathbf{x}} = \mathbf{A}(t) \cdot \mathbf{x}$, $\tilde{t} = t$, leading to the change of derivatives

$$\partial_t = \frac{\partial \tilde{x}_i}{\partial t} \tilde{\partial}_i + \frac{\partial \tilde{t}}{\partial t} \tilde{\partial}_t \tag{21}$$

$$= \partial_t [A_{ij}(t) x_j] \tilde{\partial}_i + \tilde{\partial}_t \tag{22}$$

$$= [(\partial_t \mathbf{A})\mathbf{A}^{-1}\tilde{\mathbf{x}}]_i \tilde{\partial}_i + \tilde{\partial}_t \tag{23}$$

$$\dot{C}_{ij} \partial_i \partial_j = \dot{C}_{ij} \frac{\partial \tilde{x}_k}{\partial x_i} \frac{\partial \tilde{x}_l}{\partial x_j} \tilde{\partial}_k \tilde{\partial}_l \tag{24}$$

$$= \dot{C}_{ij} A_{ki} A_{lj} \tilde{\partial}_k \tilde{\partial}_l \tag{25}$$

$$= (\mathbf{A}\dot{\mathbf{C}}\mathbf{A}^\top)_{kl} \tilde{\partial}_k \tilde{\partial}_l \tag{26}$$

and the Fokker-Planck Equation

$$\left[[(\partial_t\mathbf{A})\mathbf{A}^{-1}\tilde{\mathbf{x}}]_i\tilde{\partial}_i + \tilde{\partial}_t\right]\tilde{p}_{\tilde{t}}(\tilde{\mathbf{x}}) = \frac{1}{2}(\mathbf{A}\dot{\mathbf{C}}\mathbf{A}^\top)_{kl}\tilde{\partial}_k\tilde{\partial}_l\tilde{p}_{\tilde{t}}(\tilde{\mathbf{x}}), \tag{27}$$

where $\tilde{p}_{\tilde{t}}(\tilde{\mathbf{x}}) = p_t(\mathbf{x})$ is simply written in rescaled coordinates. However, this is not a properly normalized probability distribution in the *rescaled* coordinates, so we define $q(\tilde{\mathbf{x}}, \tilde{t}) \equiv J^{-1}(\tilde{t})\tilde{p}_{\tilde{t}}(\tilde{\mathbf{x}})$, which in turn satisfies

$$\left[[(\partial_t\mathbf{A})\mathbf{A}^{-1}\tilde{\mathbf{x}}]_i\tilde{\partial}_i + \tilde{\partial}_t + \tilde{\partial}_t\log J\right]q(\tilde{\mathbf{x}}, \tilde{t}) = \frac{1}{2}(\mathbf{A}\dot{\mathbf{C}}\mathbf{A}^\top)_{kl}\tilde{\partial}_k\tilde{\partial}_l q(\tilde{\mathbf{x}}, \tilde{t}). \tag{28}$$

Now consider the time-dependent Gaussian density

$$q(\tilde{\mathbf{x}}, \tilde{t}) = \frac{1}{\sqrt{(2\pi)^{\frac{d}{2}}|\mathbf{\Sigma}||\mathbf{A}^\top\mathbf{A}|}}\exp\left(-\frac{1}{2}(\tilde{\mathbf{x}} - \mathbf{A}\boldsymbol{\mu})^\top(\mathbf{A}\mathbf{\Sigma}\mathbf{A}^\top)^{-1}(\tilde{\mathbf{x}} - \mathbf{A}\boldsymbol{\mu})\right) \tag{29}$$

with rescaling factor $J(\tilde{t}) = |\mathbf{A}^\top\mathbf{A}(t)|$. We then calculate the pieces of (28) as follows:

$$\tilde{\nabla}q = -(\mathbf{A}\mathbf{\Sigma}\mathbf{A}^\top)^{-1}(\tilde{\mathbf{x}} - \mathbf{A}\boldsymbol{\mu})q$$

$$\tilde{\partial}_i\tilde{\partial}_j q = \left[(\mathbf{A}\mathbf{\Sigma}\mathbf{A}^\top)^{-1}(\tilde{\mathbf{x}} - \mathbf{A}\boldsymbol{\mu})\right]_i\left[(\mathbf{A}\mathbf{\Sigma}\mathbf{A}^\top)^{-1}(\tilde{\mathbf{x}} - \mathbf{A}\boldsymbol{\mu})\right]_j q - [(\mathbf{A}\mathbf{\Sigma}\mathbf{A}^\top)^{-1}]_{ij}q$$

$$\tilde{\partial}_t\log J = \tilde{\partial}_t\log|\mathbf{A}\mathbf{A}^\top| = \mathrm{tr}(\tilde{\partial}_t\log\mathbf{A}\mathbf{A}^\top) = \mathrm{tr}\left((\mathbf{A}\mathbf{A}^\top)^{-1}\left[(\tilde{\partial}_t\mathbf{A})\mathbf{A}^\top + \mathbf{A}(\tilde{\partial}_t\mathbf{A}^\top)\right]\right)$$

$$\begin{aligned}
\tilde{\partial}_t q = &-\frac{1}{2}\mathrm{tr}((\mathbf{A}\mathbf{\Sigma}\mathbf{A}^\top)^{-1}\tilde{\partial}_t(\mathbf{A}\mathbf{\Sigma}\mathbf{A}^\top))q \\
&+ q\boldsymbol{\mu}^\top\tilde{\partial}_t\mathbf{A}^\top(\mathbf{A}\mathbf{\Sigma}\mathbf{A}^\top)^{-1}(\tilde{\mathbf{x}} - \mathbf{A}\boldsymbol{\mu}) \\
&- \frac{q}{2}\mathrm{tr}\left[(\tilde{\mathbf{x}} - \mathbf{A}\boldsymbol{\mu})(\tilde{\mathbf{x}} - \mathbf{A}\boldsymbol{\mu})^\top\tilde{\partial}_t(\mathbf{A}\mathbf{\Sigma}\mathbf{A}^\top)^{-1}\right] \\
&- \tilde{\partial}_t\log J
\end{aligned}$$

$$\begin{aligned}
\tilde{\partial}_t(\mathbf{A}\mathbf{\Sigma}\mathbf{A}^\top)^{-1} = &-(\mathbf{A}\mathbf{\Sigma}\mathbf{A}^\top)^{-1}\tilde{\partial}_t(\mathbf{A}\mathbf{\Sigma}\mathbf{A}^\top)(\mathbf{A}\mathbf{\Sigma}\mathbf{A}^\top)^{-1} \\
= &-(\mathbf{A}\mathbf{\Sigma}\mathbf{A}^\top)^{-1}((\tilde{\partial}_t\mathbf{A})\mathbf{A}^{-1}) - ((\tilde{\partial}_t\mathbf{A})\mathbf{A}^{-1})^\top(\mathbf{A}\mathbf{\Sigma}\mathbf{A}^\top)^{-1} \\
&- \mathbf{A}^{-\top}\mathbf{\Sigma}^{-1}\tilde{\partial}_t\mathbf{\Sigma}\mathbf{\Sigma}^{-1}\mathbf{A}^{-1}.
\end{aligned}$$

With these results, the left and right sides of (28) become

$$\begin{aligned}
{\color{red}[\tilde{\mathbf{x}}^\top \cdot \tilde{\partial}_t\log\mathbf{A}^\top \cdot \tilde{\nabla}} + \tilde{\partial}_t + {\color{orange}\tilde{\partial}_t\log J}]q = &-\tilde{\mathbf{x}}^\top[(\tilde{\partial}_t\mathbf{A})\mathbf{A}^{-1}]^\top(\mathbf{A}\mathbf{\Sigma}\mathbf{A}^\top)^{-1}(\tilde{\mathbf{x}} - \mathbf{A}\boldsymbol{\mu})q \\
&- \frac{1}{2}\mathrm{tr}((\mathbf{A}\mathbf{\Sigma}\mathbf{A}^\top)^{-1}\tilde{\partial}_t(\mathbf{A}\mathbf{\Sigma}\mathbf{A}^\top))q \\
&+ \boldsymbol{\mu}^\top\tilde{\partial}_t\mathbf{A}^\top(\mathbf{A}\mathbf{\Sigma}\mathbf{A}^\top)^{-1}(\tilde{\mathbf{x}} - \mathbf{A}\boldsymbol{\mu})q \\
&- \frac{1}{2}\mathrm{tr}\left((\tilde{\mathbf{x}} - \mathbf{A}\boldsymbol{\mu})(\tilde{\mathbf{x}} - \mathbf{A}\boldsymbol{\mu})^\top\tilde{\partial}_t(\mathbf{A}\mathbf{\Sigma}\mathbf{A}^\top)^{-1}\right)q \\
&- \tilde{\partial}_t\log|\mathbf{A}\mathbf{A}^\top|q \\
&+ \mathrm{tr}\left((\mathbf{A}\mathbf{A}^\top)^{-1}\left[(\tilde{\partial}_t\mathbf{A})\mathbf{A}^\top + \mathbf{A}(\tilde{\partial}_t\mathbf{A}^\top)\right]\right)q \\
= &-\frac{q}{2}\mathrm{tr}\left(\tilde{\partial}_t(\mathbf{A}\mathbf{\Sigma}\mathbf{A}^\top)(\mathbf{A}\mathbf{\Sigma}\mathbf{A}^\top)^{-1}\right) \\
&+ \frac{q}{2}\mathrm{tr}\left((\mathbf{A}\mathbf{\Sigma}\mathbf{A}^\top)^{-1}(\tilde{\mathbf{x}} - \mathbf{A}\boldsymbol{\mu})(\tilde{\mathbf{x}} - \mathbf{A}\boldsymbol{\mu})^\top\left[\tilde{\partial}_t(\mathbf{A}\mathbf{\Sigma}\mathbf{A}^\top)(\mathbf{A}\mathbf{\Sigma}\mathbf{A}^\top)^{-1}\right]\right)
\end{aligned}$$

$$\begin{aligned}
(\mathbf{A}\dot{\mathbf{C}}\mathbf{A}^\top)_{kl}\tilde{\partial}_k\tilde{\partial}_l q = &-\mathrm{tr}(\mathbf{A}\dot{\mathbf{C}}\mathbf{A}^\top(\mathbf{A}\mathbf{\Sigma}\mathbf{A}^\top)^{-1})q \\
&+ \mathrm{tr}\left((\tilde{\mathbf{x}} - \mathbf{A}\boldsymbol{\mu})^\top(\mathbf{A}\mathbf{\Sigma}\mathbf{A}^\top)^{-1}(\mathbf{A}\dot{\mathbf{C}}\mathbf{A}^\top)(\mathbf{A}\mathbf{\Sigma}\mathbf{A}^\top)^{-1}(\tilde{\mathbf{x}} - \mathbf{A}\boldsymbol{\mu})\right)q
\end{aligned}$$

and $q(\tilde{\mathbf{x}}, \tilde{t})$ is a solution when

$$\begin{aligned}
\frac{1}{2}\mathbf{A}\dot{\mathbf{C}}\mathbf{A}^\top(\mathbf{A}\mathbf{\Sigma}\mathbf{A}^\top)^{-1} &= \frac{1}{2}\tilde{\partial}_t(\mathbf{A}\mathbf{\Sigma}\mathbf{A}^\top)(\mathbf{A}\mathbf{\Sigma}\mathbf{A}^\top)^{-1} \\
\Rightarrow \quad \mathbf{A}\dot{\mathbf{C}}\mathbf{A}^\top &= \tilde{\partial}_t(\mathbf{A}\mathbf{\Sigma}\mathbf{A}^\top). \tag{30}
\end{aligned}$$

Thus, for $q$ to be a solution in the absence of rescaling ($\mathbf{A} = \mathbb{1}$) requires $\dot{\boldsymbol{\Sigma}} = \dot{\mathbf{C}}$, and combining this with (30) gives the additional constraint

$$\dot{\mathbf{A}}\boldsymbol{\Sigma}\mathbf{A}^\top + \mathbf{A}\boldsymbol{\Sigma}\dot{\mathbf{A}}^\top = \mathbf{0}. \tag{31}$$

Finally, note that, under the assumed form of $p_t(\mathbf{x})$ given in (5), when $\mathbf{C}(t)$ increases without bound, $q(\tilde{\mathbf{x}}, t) \to \mathcal{N}(\mathbf{0}, \mathbf{A}\mathbf{C}\mathbf{A}^\top(t))$ asymptotically (under rescaling), and this distribution is stationary when $\tilde{\boldsymbol{\Sigma}}(t) = \mathbf{A}\boldsymbol{\Sigma}\mathbf{A}^\top \to \mathbf{A}\mathbf{C}\mathbf{A}^\top$ is time-independent and a solution to (31).

### A.3 Derivation of the inflationary pfODE

Here, we derive the form of the pfODE (6) in rescaled coordinates. Starting from the unscaled inflationary process (**Appendix A.1**) with $\mathbf{f} = \mathbf{0}$ and $\mathbf{G}\mathbf{G}^\top(\mathbf{x}, t) = \dot{\mathbf{C}}(t)$, substituting into (4) gives the pfODE

$$\frac{d\mathbf{x}}{dt} = -\frac{1}{2}\dot{\mathbf{C}}(t) \cdot \nabla_{\mathbf{x}} \log p_t(\mathbf{x}) \tag{32}$$

As in **Appendix A.2**, we again consider the rescaling transformation $\tilde{\mathbf{x}} = \mathbf{A}(t) \cdot \mathbf{x}$, $\tilde{t} = t$. To simplify the derivation, we start by parameterizing the particle trajectory using a worldline time $\tau$ such that $dt = d\tau$ while $\mathbf{A}$ remains a function of $t$. With this convention, the pfODE becomes

$$\frac{d\tilde{x}_i}{d\tau} = \frac{\partial \tilde{x}_i}{\partial x_j}\frac{dx_j}{d\tau} + \frac{\partial \tilde{x}_i}{\partial t}\frac{dt}{d\tau} \tag{33}$$

$$= A_{ij}\frac{dx_j}{d\tau} + \frac{\partial(\mathbf{A}\mathbf{x})_i}{\partial t} \tag{34}$$

$$= A_{ij}\frac{dx_j}{d\tau} + \sum_{jk} (\partial_t A_{ij}) A_{jk}^{-1} A_{kl}x_l \quad \Rightarrow \tag{35}$$

$$\frac{d\tilde{\mathbf{x}}}{d\tau} = \mathbf{A}\frac{d\mathbf{x}}{d\tau} + \left[(\partial_t \mathbf{A})\mathbf{A}^{-1}\right] \cdot \tilde{\mathbf{x}} \tag{36}$$

$$= \mathbf{A} \cdot \left(-\frac{1}{2}\dot{\mathbf{C}} \cdot \nabla_{\mathbf{x}} \log p_t(\mathbf{x})\right) + \left[(\partial_t \mathbf{A})\mathbf{A}^{-1}\right] \cdot \tilde{\mathbf{x}}. \tag{37}$$

Two important things to note about this form: First, the score function $\nabla_{\mathbf{x}} \log p_t(\mathbf{x})$ is calculated in the *unscaled* coordinates. In practice, this is the form we use when integrating the pfODE, though the transformation to the scaled coordinates is straightforward. Second, the rescaling has induced a second force due to the change of measure factor, and this force points inward toward the origin when $\mathbf{A}$ is a contraction. This overall attraction thus balances the repulsion from areas of high local density due to the negative score function, with the result that the asymptotic distribution is stabilized.

More formally, recalling the comments at the conclusion of **Appendix A.2**, when $\mathbf{C}(t)$ grows without bound in (5), $p_t(\mathbf{x})$, the unscaled density, is asymptotically Gaussian with covariance $\mathbf{C}(t)$, and its rescaled form $q(\tilde{\mathbf{x}}, \tilde{t})$ is a stationary solution of the corresponding rescaled Fokker-Planck Equation. In this case, we also have

$$\frac{d\tilde{\mathbf{x}}}{d\tau} \xrightarrow[t\to\infty]{} \left(\frac{1}{2}\mathbf{A}\dot{\mathbf{C}}\mathbf{C}^{-1} + \dot{\mathbf{A}}\right) \cdot \mathbf{x} = \mathbf{0}, \tag{38}$$

where we have made use of (31) with $\boldsymbol{\Sigma} \to \mathbf{C}$. That is, when the rescaling and flow are chosen such that the (rescaled) diffusion PDE has a stationary Gaussian solution, points on the (rescaled) flow ODE eventually stop moving.

### A.4 Equivalence of inflationary flows and standard pfODEs

Here, we show that our pfODE in (6) is equivalent to the form proposed by [49] for isotropic $\mathbf{C}(t)$ and $\mathbf{A}(t)$. We begin by taking equation (6) and rewriting it such that our score term is computed with respect to the rescaled variable $\tilde{\mathbf{x}}$:

$$\frac{d\tilde{\mathbf{x}}}{d\tilde{t}} = \mathbf{A} \cdot \left(-\frac{1}{2}\dot{\mathbf{C}} \cdot \mathbf{A}^\top \cdot \mathbf{s}_{\tilde{\mathbf{x}}}(\mathbf{A}^{-1}\tilde{\mathbf{x}}, \tilde{t})\right) + \left[(\partial_t \mathbf{A})\mathbf{A}^{-1}\right] \cdot \tilde{\mathbf{x}}, \tag{39}$$

where we have made use of the transformation properties of the score function under the rescaling.

If we then choose $\mathbf{C}(t) = c^2(t)\mathbb{1}$ and $\mathbf{A}(t) = \alpha(t)\mathbb{1}$ (i.e., isotropic noising and scaling schedules), this becomes

$$\frac{\mathrm{d}\mathbf{x}}{\mathrm{d}t} = -\alpha(t)^2 \dot{c}(t)c(t)\nabla_{\mathbf{x}}\log p\left(\frac{\mathbf{x}}{\alpha(t)};t\right) + \frac{\dot{\alpha}(t)}{\alpha(t)}\mathbf{x}, \tag{40}$$

where we have dropped tildes on $\mathbf{x}$ and $t$. This is exactly the same as the form given in Equation 4 of [49] if we substitute $\alpha(t) \to s(t)$, $c(t) \to \sigma(t)$.

## A.5  Equivalence of inflationary flows and flow matching

Here, we show the equivalence of our proposed un-scaled (32) and scaled (37) pfODEs to the un-scaled and scaled ODEs obtained using the "Gaussian paths" flow matching formulation from [36]. Here, we will use the convention of the flow-matching literature in which $t = 0$ corresponds to the easily sampled distribution (e.g., Gaussian), while $t = 1$ corresponds to the target (data) distribution. In this setup, the flow $\mathbf{x}_t = \boldsymbol{\psi}_t(\mathbf{x}_0)$ is likewise specified by an ODE:

$$\frac{\mathrm{d}}{\mathrm{d}t}\boldsymbol{\psi}_t(\mathbf{x}_0) = \mathbf{v}_t(\boldsymbol{\psi}_t(\mathbf{x}_0)|\mathbf{x}_1), \tag{41}$$

where again, $\mathbf{x}_1$ is a point in the data distribution and $\mathbf{x}_0 \sim \mathcal{N}(\mathbf{0}, \mathbb{1})$. In [36], the authors show that choosing

$$\mathbf{v}_t(\mathbf{x}|\mathbf{x}_1) = \frac{\dot{\sigma}_t(\mathbf{x}_1)}{\sigma_t(\mathbf{x}_1)}(\mathbf{x} - \boldsymbol{\mu}_t(\mathbf{x}_1)) + \dot{\boldsymbol{\mu}}_t(\mathbf{x}_1) \tag{42}$$

with "dots" denoting time derivatives leads to a flow

$$\boldsymbol{\psi}_t(\mathbf{x}_0) = \sigma_t(\mathbf{x}_1)\mathbf{x}_0 + \boldsymbol{\mu}_t(\mathbf{x}_1), \tag{43}$$

that is, a conditionally linear transformation of the Gaussian sample $\mathbf{x}_0$.

For our purposes, we can re-derive (42) for the general case where $\sigma_t(\mathbf{x}_1)$ is no longer a scalar but a matrix-valued function of $\mathbf{x}_1$ and time. That is, we rewrite (43) (equation 11 in [36]) with a full covariance matrix $\boldsymbol{\Sigma}_t(\mathbf{x}_1)$:

$$\mathbf{x}_t = \boldsymbol{\psi}_t(\mathbf{x}_0) = \boldsymbol{\Sigma}_t^{\frac{1}{2}}(\mathbf{x}_1) \cdot \mathbf{x}_0 + \boldsymbol{\mu}_t(\mathbf{x}_1). \tag{44}$$

Similarly, we can write

$$\mathbf{v}_t(\mathbf{x}|\mathbf{x}_1) = \dot{\boldsymbol{\Sigma}}_t^{\frac{1}{2}}(\mathbf{x}_1)\boldsymbol{\Sigma}_t^{-\frac{1}{2}}(\mathbf{x}_1) \cdot (\mathbf{x} - \boldsymbol{\mu}_t(\mathbf{x}_1)) + \dot{\boldsymbol{\mu}}_t(\mathbf{x_1}), \tag{45}$$

from which it is straightforward to show that (41) is again satisfied.

This can be related to our pfODE (6) as follows: First, recall that, under the inflationary assumption (5) plus rescaling, our time-dependent *conditional* marginals are

$$p(\mathbf{x}_t|\mathbf{x}_1) = \mathcal{N}(\mathbf{A}_t \cdot \mathbf{x}_1, \mathbf{A}_t\mathbf{C}_t\mathbf{A}_t^\top), \tag{46}$$

which is equivalent to (44) with $\boldsymbol{\mu}_t(\mathbf{x}_1) = \mathbf{A}_t \cdot \mathbf{x}_1$, $\boldsymbol{\Sigma}_t(\mathbf{x}_1) = \mathbf{A}_t\mathbf{C}_t\mathbf{A}_t^\top$. Note that, here again, we have reversed our time conventions from the main paper to follow the flow-matching literature: $t = 0$ is our inflated Gaussian and $t = 1$ is the data distribution. From these results, along with the constraint (31) required for inflationary flows to produce a stationary Gaussian solution asymptotically, we then have, substituting into (45):

$$\dot{\boldsymbol{\Sigma}}_t^{\frac{1}{2}}\boldsymbol{\Sigma}_t^{-\frac{1}{2}} = \dot{\boldsymbol{\Sigma}}_t^{\frac{1}{2}}\boldsymbol{\Sigma}_t^{\frac{1}{2}}\boldsymbol{\Sigma}_t^{-1} = \frac{1}{2}\dot{\boldsymbol{\Sigma}}_t\boldsymbol{\Sigma}_t^{-1} \tag{47}$$

$$= \frac{1}{2}\mathbf{A}_t\dot{\mathbf{C}}_t\mathbf{A}_t^\top\boldsymbol{\Sigma}_t^{-1} \tag{48}$$

$$\Rightarrow \quad \dot{\mathbf{x}}_t = \mathbf{v}_t(\mathbf{x}_t|\mathbf{x}_1) = \frac{1}{2}\mathbf{A}_t\dot{\mathbf{C}}_t\mathbf{A}_t^\top\boldsymbol{\Sigma}_t^{-1} \cdot (\mathbf{x}_t - \mathbf{A}_t \cdot \mathbf{x}_1) + \dot{\mathbf{A}}_t \cdot \mathbf{x}_1 \tag{49}$$

$$= -\frac{1}{2}\mathbf{A}_t\dot{\mathbf{C}}_t\mathbf{A}_t^\top \cdot \nabla_{\mathbf{x}_t}\log p(\mathbf{x}_t|\mathbf{x}_1) + \dot{\mathbf{A}}_t\mathbf{A}^{-1} \cdot \mathbf{x}_t, \tag{50}$$

which is the pfODE (6) written in the rescaled form (39). Thus, our inflationary flows are equivalent to a Gaussian paths flow matching approach for a particular choice of (matrix-valued) noise schedule and mean.

### A.6 Derivation of dimension-preserving criterion

Here, for simplicity of notation, denote the participation ratio (7) by $R(\boldsymbol{\Sigma})$ and let $\boldsymbol{\Sigma} = \mathrm{diag}(\boldsymbol{\gamma})$ in its eigenbasis, so that

$$R(\boldsymbol{\gamma}) = \frac{\left(\sum_i \gamma_i\right)^2}{\sum_j \gamma_j^2} \tag{51}$$

and the change in PR under a change in covariance is given by

$$\mathrm{d}R(\boldsymbol{\gamma}) = 2\frac{\sum_i \gamma_i}{\sum_j \gamma_j^2} \sum_k \mathrm{d}\gamma_k - \frac{\left(\sum_i \gamma_i\right)^2}{\left(\sum_j \gamma_j^2\right)^2} \sum_k \gamma_k \mathrm{d}\gamma_k \tag{52}$$

$$= 2\frac{\sum_i \gamma_i}{\sum_i \gamma_i^2} \left(\mathbf{1} - R(\boldsymbol{\gamma})\frac{\boldsymbol{\gamma}}{\sum_i \gamma_i}\right) \cdot \mathrm{d}\boldsymbol{\gamma}. \tag{53}$$

Requiring that PR be preserved ($\mathrm{d}R = 0$) then gives (8).

Now, we would like to consider conditions under which PR is not preserved (i.e., (8) does not hold). Assume we are given $\dot{\boldsymbol{\gamma}}(t)$ (along with initial conditions $\boldsymbol{\gamma}(0)$) and define

$$\mathcal{R}(t) \equiv \frac{\left(\sum_i \gamma_i\right)\left(\sum_j \dot{\gamma}_j\right)}{\sum_k \gamma_k \dot{\gamma}_k} \tag{54}$$

so that

$$\left(\mathbf{1} - \mathcal{R}(t)\frac{\boldsymbol{\gamma}}{\sum_i \gamma_i}\right) \cdot \dot{\boldsymbol{\gamma}} = 0 \tag{55}$$

by definition. Then we can rewrite (8) as

$$\frac{\mathrm{d}R(\boldsymbol{\gamma})}{\mathrm{d}t} = 2\frac{\sum_i \gamma_i}{\sum_i \gamma_i^2}\left(\mathbf{1} - \mathcal{R}(t)\frac{\boldsymbol{\gamma}}{\sum_i \gamma_i}\right) \cdot \dot{\boldsymbol{\gamma}} - 2(R(\boldsymbol{\gamma}) - \mathcal{R}(t))\frac{\boldsymbol{\gamma}}{\sum_i \gamma_i^2} \cdot \dot{\boldsymbol{\gamma}}$$

$$= 0 - (R(\boldsymbol{\gamma}) - \mathcal{R}(t))\frac{\mathrm{d}}{\mathrm{d}t}\left(\log \sum_i \gamma_i^2\right)$$

$$= -(R(\boldsymbol{\gamma}) - \mathcal{R}(t))\frac{\mathrm{d}}{\mathrm{d}t}\left(\log \mathrm{Tr}(\mathbf{C}^2)\right). \tag{56}$$

In the cases we consider, flows are *expansive* ($\mathrm{d}(\log \mathrm{Tr}(\mathbf{C^2})) > 0$), with the result that (56) drives $R(\boldsymbol{\gamma})$ toward $\mathcal{R}(t)$. Thus, in cases where $\mathcal{R}(t)$ has an asymptotic value, the $R(\boldsymbol{\gamma})$ should approach this value as well. In particular, for our dimension-reducing flows, we have $\boldsymbol{\gamma} = \rho \mathbf{g} \odot \boldsymbol{\gamma}$, giving

$$\mathcal{R}(t) = \frac{\left(\sum_i \gamma_i\right)\left(\rho \sum_j g_j \gamma_j\right)}{\rho \sum_k g_k \gamma_k^2} \xrightarrow[t \to \infty]{} \frac{\left(\sum_{i=1}^K \gamma_{0i}\right)^2}{\sum_{k=1}^K \gamma_{0k}^2}, \tag{57}$$

where $i = 1 \ldots K$ are the dimensions with $g_i = g_*$ and $\gamma_k(0) = \gamma_{0k}$. That is, the asymptotic value of $\mathcal{R}(t)$ (and thus the asymptotic value of PR) is that of the covariance in which only the eigendimensions with $g_k = g_*$ have been retained, as in (10).

# B Appendix: Additional Details on Model Training and Experiments

## B.1 Derivation of Training preconditioning Terms

Following an extensive set of experiments, the authors of [49] propose a set of preconditioning factors for improving the efficiency of denoiser training (11) that forms the core of score estimation. More specifically, they parameterize the denoiser network $\mathbf{D}_\theta(\mathbf{x}; \sigma)$ as

$$\mathbf{D}_\theta(\mathbf{x}, \sigma) = c_{skip}(\sigma)\mathbf{x} + c_{out}(\sigma)\mathbf{F}_\theta(c_{in}(\sigma)\mathbf{x}; c_{noise}(\sigma)), \tag{58}$$

where $F_\theta$ is the actual neural network being trained and $c_{in}, c_{out}, c_{skip}$, and $c_{noise}$ are preconditioning factors. Using this parameterization of $\mathbf{D}_\theta(\mathbf{x}; \sigma)$, they then re-write the original $L_2$ de-noising loss as

$$\mathcal{L}(\mathbf{D}_\theta) = \mathbb{E}_{\sigma, \mathbf{y}, \mathbf{n}} \left[ w(\sigma) \| \mathbf{F}_\theta(c_{in} \cdot (\mathbf{y} + \mathbf{n}); c_{noise}(\sigma)) - \frac{1}{c_{out}} (\mathbf{y} - c_{skip}(\sigma) \cdot (\mathbf{y} + \mathbf{n})) \|_2^2 \right], \tag{59}$$

where $w(\sigma)$ is also a preconditioning factor, $\mathbf{y}$ is the original data sample, $\mathbf{n}$ is a noise sample and $\mathbf{x} = \mathbf{y} + \mathbf{n}$. As detailed in [49], these "factors" stabilize DBM training by:

1. $c_{in}$: Scaling inputs to unit variance across all dimensions, and for all noise/perturbation levels. This is essential for stable neural net training via gradient descent.

2. $c_{out}$: Scaling the effective network output to unit variance across dimensions.

3. $c_{skip}$: Compensating for $c_{out}$, thus ensuring network errors are minimally amplified. The authors of [49] point out that this factor allows the network to choose whether to predict the target, its residual, or some value between the two.

4. $w(\sigma)$: Uniformizing the weight given to different noise levels in the total loss.

5. $c_{noise}$: Determining how noise levels should be sampled during training so that the trained network efficiently covers different noise levels. This is the conditioning noise input fed to the network along with the perturbed data. This quantity is determined empirically.

In [49], the authors propose optimal forms for all of these quantities based on these plausible first principles (cf. Table 1 and Appendix B.6 of that work). However, the forms proposed there rely strongly on the assumption that the noise schedule is isotropic, which does not hold for our inflationary schedules, which are diagonal but not proportional to the identity. Here, we derive analogous expressions for our setting.

As in the text, assume we work in the eigenbasis of the initial data distribution $\mathbf{\Sigma_0}$ and let $\mathbf{C}(t) = \mathrm{diag}(\boldsymbol{\gamma}(t))$ be the noising schedule, such that the data covariance at time $t$ is $\mathbf{\Sigma}(t) = \mathbf{\Sigma_0} + \mathbf{C}(t)$. Assuming a noise-dependent weighting factor $\mathbf{\Lambda}(t)$ analogous to $\sqrt{w(\sigma)}$ above, we then rewrite (11) as

$$\mathcal{L}(\mathbf{D}_\theta) = \mathbb{E}_{\mathbf{t}, \mathbf{y}, \mathbf{n}} \left[ \| \mathbf{\Lambda}(t)(\mathbf{D}_\theta(\mathbf{y} + \mathbf{n}; \boldsymbol{\gamma}(t)) - \mathbf{y}) \|^2 \right] \tag{60}$$

$$= \mathbb{E}_{\mathbf{t}, \mathbf{y}, \mathbf{n}} \left[ \| \mathbf{\Lambda}(t) \left( \mathbf{C_{out}} \mathbf{F}_\theta(\mathbf{C_{in}}(\mathbf{y} + \mathbf{n}); \mathbf{c_{noise}}) - (\mathbf{y} - \mathbf{C_{skip}}(\mathbf{y} + \mathbf{n})) \right) \|^2 \right] \tag{61}$$

$$= \mathbb{E}_{\mathbf{t}, \mathbf{y}, \mathbf{n}} \left[ \| \mathbf{\Lambda}(t) \mathbf{C_{out}} \left( \mathbf{F}_\theta(\mathbf{C_{in}}(\mathbf{y} + \mathbf{n}); \mathbf{c_{noise}}) - \mathbf{C_{out}^{-1}}(\mathbf{y} - \mathbf{C_{skip}}(\mathbf{y} + \mathbf{n})) \right) \|^2 \right] \tag{62}$$

This clearly generalizes (59) by promoting all preconditioning factors either to matrices $(\mathbf{C_{in}}, \mathbf{C_{out}}, \mathbf{C_{skip}}, \mathbf{\Lambda})$ or vectors $(\mathbf{c_{noise}})$. We now derive forms for each of these preconditioning factors.

### B.1.1 $\mathbf{C_{in}}$

The goal is to choose $\mathbf{C_{in}}$ such that its application to the noised input $\mathbf{y} + \mathbf{n}$ has unit covariance:

$$\mathbb{1} = \mathrm{Var}_{\mathbf{y}, \mathbf{n}} \left[ \mathbf{C_{in}}(\mathbf{y} + \mathbf{n}) \right] \tag{63}$$

$$= \mathbf{C_{in}} \mathrm{Var}_{\mathbf{y}, \mathbf{n}} \left[ (\mathbf{y} + \mathbf{n}) \right] \mathbf{C_{in}^\top} \tag{64}$$

$$= \mathbf{C_{in}} \left( \mathbf{\Sigma_0} + \mathbf{C}(t) \right) \mathbf{C_{in}^\top} \tag{65}$$

$$= \mathbf{C_{in}} \mathbf{\Sigma}(t) \mathbf{C_{in}^\top} \tag{66}$$

$$\Rightarrow \quad \mathbf{C_{in}} = \mathbf{\Sigma}^{-\frac{1}{2}}(t) \tag{67}$$

More explicitly, if $\mathbf{W}$ is the matrix whose columns are the eigenvectors of $\mathbf{\Sigma_0}$, then

$$\mathbf{C_{in}} = \mathbf{W}\mathrm{diag}\left(1/\sqrt{\sigma_0^2 + \gamma(t)}\right)\mathbf{W}^\top, \tag{68}$$

where the square root is taken elementwise.

### B.1.2  $\mathbf{C_{out}}$, $\mathbf{C_{skip}}$

We begin by imposing the requirement that the target for the neural network $\mathbf{F}$ should have identity covariance:

$$\mathbb{1} = \mathrm{Var}_{\mathbf{y},\mathbf{n}}\left[\mathbf{C_{out}}^{-1}(\mathbf{y} - \mathbf{C_{skip}}(\mathbf{y} + \mathbf{n}))\right] \tag{69}$$

$$\Rightarrow \quad \mathbf{C_{out}}\mathbf{C_{out}}^\top = \mathrm{Var}_{\mathbf{y},\mathbf{n}}\left[\mathbf{y} - \mathbf{C_{skip}}(\mathbf{y} + \mathbf{n})\right]$$

$$= \mathrm{Var}_{\mathbf{y},\mathbf{n}}\left[(\mathbb{1} - \mathbf{C_{skip}})\mathbf{y} - \mathbf{C_{skip}}\mathbf{n}\right]$$

$$= (\mathbb{1} - \mathbf{C_{skip}})\mathbf{\Sigma_0}(\mathbb{1} - \mathbf{C_{skip}})^\top + \mathbf{C_{skip}}\mathbf{C}(t)\mathbf{C_{skip}}^\top. \tag{70}$$

This generalizes Equation 123 in Appendix B.6 of [49].

Again by analogy with [49], we choose $\mathbf{C_{skip}}$ to minimize the left-hand side of (70):

$$\mathbf{0} = -(\mathbb{1} - \mathbf{C_{skip}})\mathbf{\Sigma_0} + \mathbf{C_{skip}}\mathbf{C}(t) \tag{71}$$

$$\Rightarrow \quad \mathbf{\Sigma_0} = \mathbf{C_{skip}}\mathbf{\Sigma}(t) \tag{72}$$

$$\Rightarrow \quad \mathbf{C_{skip}} = \mathbf{\Sigma_0}\mathbf{\Sigma^{-1}}(t) = \mathbf{W}\mathrm{diag}\left(\sigma_0^2/\left(\sigma_0^2 + \gamma(t)\right)\right)\mathbf{W}^\top, \tag{73}$$

which corresponds to Equation 131 in Appendix B.6 of [49].

Using (73) in (70) then allows us to solve for $\mathbf{C_{out}}$:

$$\mathbf{C_{out}}\mathbf{C_{out}}^\top = (\mathbb{1} - \mathbf{\Sigma_0}\mathbf{\Sigma}^{-1})\mathbf{\Sigma_0}(\mathbb{1} - \mathbf{\Sigma_0}\mathbf{\Sigma}^{-1})^\top + \mathbf{\Sigma_0}\mathbf{\Sigma}^{-1}\mathbf{C}\mathbf{\Sigma}^{-1}\mathbf{\Sigma_0} \tag{74}$$

$$= \mathbf{\Sigma_0} - 2\mathbf{\Sigma_0}\mathbf{\Sigma}^{-1}\mathbf{\Sigma_0} + \mathbf{\Sigma_0}\mathbf{\Sigma}^{-1}(\mathbf{\Sigma_0} + \mathbf{C})\mathbf{\Sigma}^{-1}\mathbf{\Sigma_0} \tag{75}$$

$$= \mathbf{\Sigma_0} - \mathbf{\Sigma_0}\mathbf{\Sigma}^{-1}\mathbf{\Sigma_0} \tag{76}$$

$$= \left(\mathbf{\Sigma_0^{-1}} + \mathbf{C^{-1}}(t)\right)^{-1} \tag{77}$$

$$\Rightarrow \quad \mathbf{C_{out}} = \mathbf{W}\mathrm{diag}\left(\sqrt{\sigma_0^2 \odot \gamma(t)/\left(\sigma_0^2 + \gamma(t)\right)}\right)\mathbf{W}^\top \tag{78}$$

### B.1.3  $\mathbf{\Lambda}(t)$

Our goal in choosing $\mathbf{\Lambda}(t)$ is to equalize the loss across different noise levels (which correspond, via the noise schedule, to different times). Looking at the form of (62), we can see that this will be satisfied when $\mathbf{\Lambda}(t)$ is chosen to cancel the outermost factor of $\mathbf{C_{out}}$

$$\mathbf{\Lambda}(t) = \mathbf{C_{out}^{-1}} = \mathbf{\Sigma_0^{-1}} + \mathbf{C^{-1}}(t) = \mathbf{W}\mathrm{diag}\left(\sqrt{\sigma_0^{-2} + \gamma^{-1}(t)}\right)\mathbf{W}^\top \tag{79}$$

### B.1.4  Re-writing loss with optimal preconditioning factors

Using these results, we now rewrite (62) using the preconditioning factors derived above:

$$\mathcal{L}(\mathbf{D_\theta}) = \mathbb{E}_{\mathbf{t},\mathbf{y},\mathbf{n}}\left[\|\mathbf{\Lambda}(t)\mathbf{C_{out}}\left(\mathbf{F_\theta}(\mathbf{C_{in}}(\mathbf{y} + \mathbf{n}); \mathbf{c_{noise}}) - \mathbf{C_{out}^{-1}}(\mathbf{y} - \mathbf{C_{skip}}(\mathbf{y} + \mathbf{n})))\|^2\right]$$

$$= \mathbb{E}_{\mathbf{t},\mathbf{y},\mathbf{n}}\left[\|\mathbf{F_\theta}(\mathbf{\Sigma}^{-\frac{1}{2}}(t)\cdot(\mathbf{y} + \mathbf{n}); \mathbf{c_{noise}}) - \left(\mathbf{\Sigma_0^{-1}} + \mathbf{C^{-1}}(t)\right)^{\frac{1}{2}}\cdot(\mathbf{y} - \mathbf{\Sigma_0}\mathbf{\Sigma}^{-1}(t)\cdot(\mathbf{y} + \mathbf{n}))\|^2\right].$$

In practice, we precompute $\mathbf{W}$ and $\sigma_0^2$ via SVD and compute all relevant precoditioners in eigenspace using the forms given above. For $\mathbf{c_{noise}}$, we follow the same noise conditionining scheme used in the DDPM model [27], sampling $t$ uniformly from some interval $t \sim \mathcal{U}[t_{min}, t_{max}]$ and then setting $c_{noise} = (M - 1)t$, for some scalar hyperparameter $M$. We choose $M = 1000$, in agreement with [49, 27]. After this, as indicated above, our noise is sampled via $\mathbf{n} \sim \mathcal{N}(\mathbf{0}, \mathbf{C}(t))$ with $\mathbf{C}(t) = \mathbf{W}\mathrm{diag}(\gamma(t))\mathbf{W}^\top$.

## B.2 Construction of **g** and its impact on compression and generative performance of PR-Reducing pfODEs

As highlighted in main text, for constant end integration time $T$ and $\rho$, the final scale ratio between preserved and compressed dimensions is dictated by the quantity $g_* - g_i$, which we called the *inflation gap* (IG). Higher inflation gaps (IGs) lead to more stringent exponential shrinkage towards zero in compressed dimensions (**Tables 6, 11**) and worse off generative performance (**Table 2**).

In PR-Reducing experiments, we set $\rho = 1$ and constructed **g** by making all elements of **g** corresponding to preserved dimensions equal to 2 (i.e., $g_{preserved} = g_{max} = 2$) and all elements corresponding to compressed dimensions equal to $g_{compressed} = g_{min} = g_{preserved} - $ IG (**Tables 5, 10**). Of note, for PR-Preserving experiments, all elements of **g** are set to 1 (i.e., $\mathbf{g} = \mathbf{1}$, IG $= 0$) and we chose $\rho = 2$, such that all dimensions are inflated to the same extent and we match exponential constant used for preserved dimensions in PR-Reducing experiments.

## B.3 Details of pfODE integration

### B.3.1 pfODE in terms of network outputs

Here we rewrite the pfODE (6) in terms of the network outputs $\mathbf{D}(\mathbf{x}, \text{diag}(\boldsymbol{\gamma}(t)))$, learned during training and queried in our experiments. As described in **Appendix B.1.4** and in line with previous DBM training approaches, we opt to use time directly as our network conditioning input. That is, our networks are parameterized as $\mathbf{D}(\mathbf{x}, t)$. Then, using the fact that the score can be written in terms of the network as [30, 49]

$$\nabla_{\mathbf{x}} \log p(\mathbf{x}, \mathbf{C}(t)) = \mathbf{C}^{-1}(t) \cdot (\mathbf{D}(\mathbf{x}, t) - \mathbf{x}), \tag{80}$$

we rewrite (6) as

$$\frac{\mathrm{d}\tilde{\mathbf{x}}}{\mathrm{d}\tilde{t}} = -\frac{1}{2}\mathbf{A}\dot{\mathbf{C}}\left[\mathbf{C}^{-1}(\mathbf{D}(\mathbf{x}, t) - \mathbf{x})\right] + \left[(\partial_t \mathbf{A})\mathbf{A}^{-1}\right] \cdot \tilde{\mathbf{x}} \tag{81}$$

$$= -\frac{1}{2}\mathbf{A}\dot{\mathbf{C}}\left[\mathbf{C}^{-1}(\mathbf{D}(\mathbf{A}^{-1} \cdot \tilde{\mathbf{x}}, t) - \mathbf{A}^{-1} \cdot \tilde{\mathbf{x}})\right] + \left[(\partial_t \mathbf{A})\mathbf{A}^{-1}\right] \cdot \tilde{\mathbf{x}} \tag{82}$$

$$= -\frac{1}{2}\boldsymbol{\alpha}(t) \odot \frac{\dot{\boldsymbol{\gamma}}(t)}{\boldsymbol{\gamma}(t)} \odot \left(\mathbf{D}\left(\frac{\tilde{\mathbf{x}}}{\boldsymbol{\alpha}(t)}, t\right) - \frac{\tilde{\mathbf{x}}}{\boldsymbol{\alpha}(t)}\right) + \frac{\dot{\boldsymbol{\alpha}}(t)}{\boldsymbol{\alpha}(t)} \odot \tilde{\mathbf{x}}, \tag{83}$$

where in the last line we have expressed $\mathbf{A}(t)$ and $\dot{\mathbf{C}}\mathbf{C}^{-1}$ in their respective eigenspace (diagonal) representations, where the divisions are to be understood element-wise. For PR-Reducing schedules, this expression simplifies even further, since our scaling schedule becomes isotropic - i.e., $\mathbf{A}(t) = \alpha(t)\mathbb{1}$.

### B.3.2 Solvers and Discretization Schedules

To integrate (83), we utilize either Euler's method for toy datasets and Heun's method (see **Algorithm 1**) for high-dimensional image datasets. The latter has been shown to provide better tradeoffs between number of neural function evaluations (NFEs) and image quality as assessed through FID scores in larger data sets [49].

In toy data examples, we chose a simple, linearly spaced (step size $h = 10^{-2}$) discretization scheme, integrating from $t = 0$ to $t = t_{max}$ when inflating and reversing these endpoints when generating data from the latent space. For higher-dimensional image datasets (CIFAR-10, AFHQv2), we instead discretized using $t_i = \frac{i}{N-1}(t_{max} - \epsilon_s) + \epsilon_s$ when inflating, where $t_{max}$ is again the maximum time at which networks were trained to denoise and $\epsilon_s = 10^{-2}$, similar to the standard discretization scheme for VP-ODEs [49, 30] (though we do not necessarily enforce $t_{max} = 1$). When generating from latent space, this discretization is preserved but integration is performed in reverse.

## B.4 Training Details

### B.4.1 Toy DataSets

Toy models were trained using a smaller convolutional UNet architecture (*ToyConvUNet*) and our proposed preconditioning factors (**Appendix B.1**). For all toy datasets, we trained networks both

---

**Algorithm 1** Eigen-Basis pfODE Simulation using Heun's $2^{nd}$ order method

---

1: **procedure** HEUNSAMPLER($\mathbf{D}_\theta(\mathbf{x}, t)$, $\boldsymbol{\gamma}(t)$, $\boldsymbol{\alpha}(t)$, $\mathbf{W}^\top$, $t_{i \in \{0,...,N\}}$)

2:      **if** running "generation" **then**          ▷ Generate initial sample at $t_0$

3:          $\tilde{\mathbf{x}}_0 \sim \mathcal{N}(\mathbf{0}, \text{diag}(\boldsymbol{\alpha}(t_0) \odot \boldsymbol{\gamma}(t_0)))$      ▷ Sample from Gaussian latent space

4:      **else**          ▷ i.e., if running "inflation"

5:          $\mathbf{x}_0 \sim p_{data}(\mathbf{x})$      ▷ Sample from target distribution

6:          $\tilde{\mathbf{x}}_0 = \boldsymbol{\alpha}(t_0)(\mathbf{W}^\top \cdot \mathbf{x}_0)$      ▷ Transform to eigenbasis, scale

7:      **end if**

8:      **for** $i \in \{0, 1, ..., N-1\}$ **do**:      ▷ Solve equation (83) $N$ times

9:          $\tilde{\mathbf{d}}_i \leftarrow -\frac{1}{2}\boldsymbol{\alpha}(t_i) \odot \frac{\dot{\boldsymbol{\gamma}}(t_i)}{\boldsymbol{\gamma}(t_i)} \odot \left( \mathbf{D}\left( \frac{\tilde{\mathbf{x}}_i}{\boldsymbol{\alpha}(t_i)}, t_i \right) - \frac{\tilde{\mathbf{x}}_i}{\boldsymbol{\alpha}(t_i)} \right)$

10:          $+ \frac{\dot{\boldsymbol{\alpha}}(t_i)}{\boldsymbol{\alpha}(t_i)} \odot \tilde{\mathbf{x}}_i$      ▷ Evaluate $\frac{d\tilde{\mathbf{x}}}{dt}$ at $t_i$

11:          $\tilde{\mathbf{x}}_{i+1} \leftarrow \tilde{\mathbf{x}}_i + (t_{i+1} - t_i)\tilde{\mathbf{d}}_i, \quad \mathbf{x}_{i+1} = \frac{\tilde{\mathbf{x}}_{i+1}}{\boldsymbol{\alpha}(t_{i+1})}$      ▷ Take Euler step from $t_i$ to $t_{i+1}$

12:          $\tilde{\mathbf{d}}_i' \leftarrow -\frac{1}{2}\boldsymbol{\alpha}(t_{i+1}) \odot \frac{\dot{\boldsymbol{\gamma}}(t_{i+1})}{\boldsymbol{\gamma}(t_{i+1})} \odot \left( \mathbf{D}\left( \frac{\tilde{\mathbf{x}}_{i+1}}{\boldsymbol{\alpha}(t_{i+1})}, t_{i+1} \right) - \frac{\tilde{\mathbf{x}}_{i+1}}{\boldsymbol{\alpha}(t_{i+1})} \right)$

13:          $+ \frac{\dot{\boldsymbol{\alpha}}(t_{i+1})}{\boldsymbol{\alpha}(t_{i+1})} \odot \tilde{\mathbf{x}}_{i+1}$      ▷ Evaluate $\frac{d\tilde{\mathbf{x}}}{dt}$ at $t_{i+1}$

14:          $\tilde{\mathbf{x}}_{i+1} \leftarrow \tilde{\mathbf{x}}_i + (t_{i+1} - t_i)\left( \frac{1}{2}\tilde{\mathbf{d}}_i + \frac{1}{2}\tilde{\mathbf{d}}_i' \right)$      ▷ Apply trapezoidal rule at $t_{i+1}$

15:      return $\tilde{\mathbf{x}}_N$      ▷ Return Sample

16: **end procedure**

---

by using original images as inputs (i.e., "image space basis") or by first transforming images to their PCA representation (i.e., "eigenbasis"). Networks trained using either base choice were able to produce qualitatively good generated samples, across all datasets. For all cases, we used a learning rate of $10^{-5}$, batch size of 8192, and exponential moving average half-life of $50 \times 10^4$. For PR-Reducing schedules, we set $\rho = 1$ and constructed $\mathbf{g}$ as described in **Appendix B.2** (**Table 5**). The only exceptions were networks used on mesh and HMC toy experiments (**Appendices C.2.1, B.6**), where we used instead $g_{preserved} = 1.15$ across all preserved dimensions (circles, S-curve) and $g_{compressed} = 0.85$ (circles), or $g_{compressed} = 0.70$ (S-curve) - **Table 5**, $2^{nd}$ and $6^{th}$ rows. This yields a softer effective compression (i.e., smaller IGs) and is needed to avoid numerical instability in these experiments.

As explained in **Appendix B.1**, to construct our $\mathbf{c}_{noise}$ preconditioning factor, we sampled $t \sim \mathcal{U}(t_{min}, t_{max})$, with $t_{min} = 10^{-7}$ across all simulations and $t_{max}$ equal to the values shown in **Table 4**. In the same table, we also show training duration (in $10^6$ images (Mimgs), as in [49]), along with both the total number of dimensions (in the original data) and the number of dimensions preserved (in latent space) for each dataset and schedule combination. In **Table 6**, we showcase latent space (i.e., end of "inflation") compressed dimension variances achieved for the different toy PR-Reducing experiments as a function of inflation gap (IG). As expected, higher IGs lead to more stringent shrinkage of compressed dimensions in latent space.

### B.4.2 CIFAR-10 and AFHQv2 Datasets

For our image datasets (i.e., CIFAR-10 and AFHQv2), we utilized similar training hyperparameters to the ones proposed by [49] for the CIFAR-10 dataset, across all schedules explored (**Table 7**).

Table 4: Toy Data Training Hyperparameters

| Dataset | Schedule | Total Dimensions | Dimensions Kept | $t_{max}$ (s) | Duration (Mimg) |
|---|---|---|---|---|---|
| Circles | PRP | 2 | 2 | 7.01 | 6975 |
| Circles | PRR | 2 | 1 | 11.01 | 8601 |
| Sine | PRP | 2 | 2 | 7.01 | 12288 |
| Sine | PRR | 2 | 1 | 11.01 | 12288 |
| Moons | PRP | 2 | 2 | 8.01 | 6400 |
| Moons | PRR | 2 | 1 | 11.01 | 8704 |
| S Curve | PRP | 3 | 3 | 9.01 | 6144 |
| S Curve | PRR | 3 | 2 | 15. 01 | 5160 |
| Swirl | PRP | 3 | 3 | 11.01 | 8704 |
| Swirl | PRR | 3 | 2 | 15.01 | 12042 |

Table 5: $g_i$ Values for Preserved vs. Compressed Dimensions for Toy Experiments.

| Dataset | Schedule | Dimensions Kept | IG | $g_{preserved}$ | $g_{compressed}$ |
|---|---|---|---|---|---|
| Circles | PRR | 1 | 2.0 | 2.0 | 0.0 |
| Circles | PRR | 1 | 0.3 | 1.15 | 0.85 |
| Sine | PRR | 1 | 2.0 | 2.0 | 0.0 |
| Moons | PRR | 1 | 2.0 | 2.0 | 0.0 |
| S Curve | PRR | 2 | 3.0 | 2.0 | -1 |
| S Curve | PRR | 2 | 0.45 | 1.15 | 0.70 |
| Swirl | PRR | 2 | 3.0 | 2.0 | -1 |

Shown in **Tables 8, 9** are our specific choices for the exponential inflation constant ($\rho$) and training duration (in $10^6$ images - Mimgs) for the two main sets of experiments performed on image datasets, namely (1) experiments with constant inflation gap (IG=1.02) and varying the number of preserved dimensions $d$ on both datasets (**Table 8**), and (2) experiments with fixed $d$ ($d = 2$) and varying inflation gaps for the AFHQV2 dataset (**Table 9**). Here, training duration was determined for each schedule based on when computed Frechet Inception Distance (FID) scores [65] stopped improving. We also showcase in **Table 10** the specific values used for elements of **g** corresponding to preserved vs. compressed dimensions at different inflation gaps.

All networks were trained on the same DDPM++ architecture, as implemented in [49] and using our proposed preconditioning scheme and factors in the standard (e.g., image space) basis. No gradient clipping or mixed-precision training were used, and all networks were trained to perform unconditional generation. We run training in the image space basis (as opposed to in eigenbasis) because this option proved to be more stable in practice for non-toy datasets. Additionally, we estimate the eigendecomposition of the target datasets before training begins using 50K samples for CIFAR-10 and 15K samples for AFHQv2. Based on our experiments, any sample size above total number of dimensions works well for estimating the desired eigenbasis.

Table 6: Toy Experiments Compressed Dimension Variance by Inflation Gap (IG)

| Dataset | Schedule | Dimensions Kept | IG | Compressed Dimension Variance |
|---|---|---|---|---|
| Circles | PRR | 1 | 2.0 | $4 \times 10^{-7}$ |
| Circles | PRR | 1 | 0.3 | $1 \times 10^{-2}$ |
| Sine | PRR | 1 | 2.0 | $4 \times 10^{-7}$ |
| Moons | PRR | 1 | 2.0 | $4 \times 10^{-7}$ |
| S Curve | PRR | 2 | 3.0 | $2 \times 10^{-12}$ |
| S Curve | PRR | 2 | 0.45 | $2.5 \times 10^{-3}$ |
| Swirl | PRR | 2 | 3.0 | $2 \times 10^{-12}$ |

Table 7: CIFAR-10 & AFHQv2 Common Training Hyperparameters (Across All Schedules)

| Hyperparameter Name | Hyperparameter Value |
|---|---|
| Channel multiplier | 128 |
| Channels per resolution | 2-2-2 |
| Dataset x-flips | No |
| Augment Probability | 12% |
| Dropout Probability | 13% |
| Learning rate | $10^{-4}$ |
| LR Ramp-Up (Mimg) | 10 |
| EMA Half-Life (Mimg) | 0.5 |
| Batch-Size | 512 |

Table 8: Training Duration (in Mimgs) and Exponential Inflation Constant ($\rho$) for Dimension Reducing Experiments Using 1.02 Inflation Gap (IG) and Dimension Preserving Experiments (IG = 0.0)

| Dataset | Total Dimensions | Dimensions Kept | IG | Training Duration | $\rho$ |
|---|---|---|---|---|---|
| CIFAR-10 | 3072 | 1 | 1.02 | 300 | 1 |
| AFHQV2 | 3072 | 1 | 1.02 | 250 | 1 |
| CIFAR-10 | 3072 | 2 | 1.02 | 300 | 1 |
| AFHQV2 | 3072 | 2 | 1.02 | 250 | 1 |
| CIFAR-10 | 3072 | 30 | 1.02 | 300 | 1 |
| AFHQV2 | 3072 | 30 | 1.02 | 450 | 1 |
| CIFAR-10 | 3072 | 40 | 1.02 | 300 | 1 |
| CIFAR-10 | 3072 | 62 | 1.02 | 250 | 1 |
| AFHQV2 | 3072 | 62 | 1.02 | 450 | 1 |
| CIFAR-10 | 3072 | 307 | 1.02 | 300 | 1 |
| AFHQV2 | 3072 | 307 | 1.02 | 300 | 1 |
| CIFAR-10 | 3072 | 615 | 1.02 | 450 | 1 |
| AFHQV2 | 3072 | 615 | 1.02 | 450 | 1 |
| CIFAR-10 | 3072 | 1536 | 1.02 | 300 | 1 |
| AFHQV2 | 3072 | 1536 | 1.02 | 250 | 1 |
| CIFAR-10 | 3072 | 3041 | 1.02 | 300 | 1 |
| AFHQV2 | 3072 | 3041 | 1.02 | 200 | 1 |
| CIFAR-10 | 3072 | 3072 | 0.00 | 275 | 2 |
| AFHQV2 | 3072 | 3072 | 0.00 | 275 | 2 |

Times utilized to construct conditioning noise inputs to networks ($\mathbf{c}_{noise}(t)$) were uniformly sampled ($t \sim \mathcal{U}(t_{min}, t_{max})$), with $t_{min} = 10^{-7}$ and $t_{max} = 15.01$, across all experiments. For the AFHQv2 dataset, we chose to adopt a 32x32 resolution (instead of 64x64 as in [49]) due to constraints on training time and GPU availability. Therefore, for our experiments, both datasets have a total of 3072 (i.e., 3x32x32) dimensions.

Finally, training was performed in a distributed fashion using either 8 or 4 GPUs per each experiment (NVIDIA GeForce GTX TITAN X, RTX 2080) in a compute cluster setting. Generation (FID) and round-trip (MSE) experiments were performed on single GPU (NVIDIA RTX 3090, 4090, A5000, A6000). We report training duration in Mimgs and note that time needed to achieve 200Mimgs is approximately 2 days on 8GPUs (4 days on 4 GPUs) using hyperparameters shown in **Tables 7, 8, 9**. This is in agreement with previous train times reported in [49] using an 8 GPU distributed training set up.

Table 9: Training Duration (in Mimgs) and Exponential Inflation Constant ($\rho$) for AFHQv2 Experiments Using Variable Inflation Gaps (IGs)

| Total Dimensions | Dimensions Kept | IG | Training Duration | $\rho$ |
|---|---|---|---|---|
| 3072 | 2 | 1.10 | 200 | 1 |
| 3072 | 2 | 1.25 | 250 | 1 |
| 3072 | 2 | 1.35 | 250 | 1 |
| 3072 | 2 | 1.50 | 200 | 1 |

Table 10: $g_i$ Values for Preserved vs. Compressed Dimensions at Different Inflation Gaps (IGs)

| Inflation Gap (IG) | $g_{\text{preserved}}$ | $g_{\text{compressed}}$ |
|---|---|---|
| 1.02 | 2.0 | 0.98 |
| 1.10 | 2.0 | 0.90 |
| 1.25 | 2.0 | 0.75 |
| 1.35 | 2.0 | 0.65 |
| 1.50 | 2.0 | 0.50 |

## B.5 Details of Roundtrip MSE and FID calculation Experiments

### B.5.1 Roundtrip Experiments

For image datasets (CIFAR-10 and AFHQv2), we simulated full round-trips: integrating the pfODEs (6) forward in time to map original images into latent space and then backwards in time to reconstruct original samples. We run these round-trips for a set of 10K randomly sampled images, three times per each schedule investigated and compute pixel mean squared error between original and reconstructed images, averaged across the 10K samples. Values reported in **Tables 1, 2** represent mean $\pm 2$ standard deviations of pixel MSE between these three different random seeds per each condition. For pfODE integration, we used the discretization schedule and Heun solver detailed above (**Appendix B.3.2**), with $t_{max} = 15.01$, $\epsilon_s = 10^{-2}$, and $N = 118$ for all conditions.

### B.5.2 FID Experiments

For image datasets, we also computed Frechet Inception Distance (FID) scores [65] across 3 independent sets of 50K random samples, per each schedule investigated. Values reported in **Tables 1, 2** represent mean $\pm 2$ standard deviations across these 3 sets of random samples per each condition. Here again, we used the discretization scheme and solver described in **Appendix B.3.2** with $t_{max} = 15.01$, $\epsilon_s = 10^{-2}$, and $N = 256$ across all conditions. We chose $N = 256$ here (instead of 118) because this provided some reasonable trade-off between improving FID scores and reducing total compute time.

To obtain our latent space random samples $\mathbf{x}(T)$ at time $t_0 = T$ (i.e, at the start of generation) we sample from a diagonal multivariate normal with either (1) all diagonal elements being 1 (for PR-Preserving schedule) or (2) all elements corresponding to preserved dimensions being 1 and all elements corresponding to compressed dimensions being equal to the *same small value* for a given inflation gap (see **Table 11**).

Table 11: Latent Space Compressed Dimensions Variance per Inflation Gap (IG), Both Datasets

| Inflation Gap (IG) | Latent Space Compressed Dimensions Variance |
|---|---|
| 1.02 | $2.15 \times 10^{-7}$ |
| 1.10 | $6.00 \times 10^{-8}$ |
| 1.25 | $6.80 \times 10^{-9}$ |
| 1.35 | $1.50 \times 10^{-9}$ |
| 1.50 | $1.76 \times 10^{-10}$ |

For our CIFAR-10 comparison experiments against existing *injective flow* models, we used the same implementations for M-Flow [21], Rectangular Flows [22], and Canonical Manifold Flows [23] as in [23]. When training the comparison *injective flows*, we used the same hyper-parameters proposed in **Appendix G.1** of [23] for the CIFAR-10 dataset. The only difference here is that we trained models with latent dimensions equal to $d = [30, 40, 62]$. Finally, comparison FID scores reported in **Table 3** represent *best* score out of 3 independently generated sets, each with 10K samples. For our comparison models, inflation gap was fixed to IG $= 1.02$ while $d$ was varied between 30, 40, and 62 and we utilized the same training hyper-parameters reported in **Tables 7, 8**. FID scores were computed using the same discretization, solver, and general set up described above.

### B.5.3 FID and Roundtrip Integration Experiments with Additional Initialization Seeds

In the FID and roundtrip pfODE integration experiments in **Tables 1, 2** we showcased variation arising from using different seeds when constructing initial generation or roundtrip samples. Another relevant source of variability in FID and MSE scores reported arises from different parameter initializations when training our denoiser networks. To assess this source of variability, we trained networks using *three additional initialization seeds for our worse performing schedule at constant inflation gap* (IG $= 1.02$, PR-Reducing to 307 dimensions), for both datasets (AFHQv2, CIFAR-10). For each such initialization seed, we conducted similar FID and roundtrip integration MSE experiments as detailed in **Sections B.5.1, B.5.2**.

**Tables 12, 13** showcase results for these experiments on each individual seed tested (top 4 rows) and aggregated across all seeds (bottom row). For top 4 rows, values reported represent mean $\pm 2\sigma$ of scores computed across three different sets of generation/roundtrip seeds (as in **Tables 1, 2**). For bottom rows, values reported represent mean $\pm 2\sigma$ of scores computed across all initialization and generation/roundtrip seeds (i.e., these represent mean $\pm 2\sigma$ over all aggregated experiments conducted for the given schedule and dataset).

Table 12: AFHQv2, FID and Roundtrip Experiments with Additional Initialization Seeds

| Seed | FID | MSE |
|---|---|---|
| 1 | $16.41 \pm 0.10$ | $3.24 \pm 0.16$ |
| 2 | $15.63 \pm 0.17$ | $3.20 \pm 0.15$ |
| 3 | $16.53 \pm 0.11$ | $3.06 \pm 0.21$ |
| 4 | $15.64 \pm 0.10$ | $3.33 \pm 0.13$ |
| all (aggregated) | $16.05 \pm 0.85$ | $3.21 \pm 0.26$ |

Table 13: CIFAR-10, FID and Roundtrip Experiments with Additional Initialization Seeds

| Seed | FID | MSE |
|---|---|---|
| 1 | $26.32 \pm 0.07$ | $1.04 \pm 0.09$ |
| 2 | $27.43 \pm 0.15$ | $0.97 \pm 0.08$ |
| 3 | $27.80 \pm 0.19$ | $0.84 \pm 0.04$ |
| 4 | $28.07 \pm 0.13$ | $0.71 \pm 0.02$ |
| all (aggregated) | $27.41 \pm 1.34$ | $0.89 \pm 0.26$ |

### B.5.4 Additional Figures for FID and Round-Trip MSE Experiments on Image Benchmark Datasets.

Below we showcase additional examples of generated images and results of roundtrip pfODE integration for different schedules explored in our experiments. More specifically, **Figure 6** showcases results of FID and roundtrip integration experiments on CIFAR-10 at constant inflation gap (IG $= 1.02$) and varying number of preserved dimensions. **Figure 7** showcases results of FID and roundtrip integration experiments done on the AFHQv2 dataset, with varying inflation gaps IG $= [1.10, 1.25, 1.35, 1.50]$ and constant number of preserved dimesions ($d = 2$).

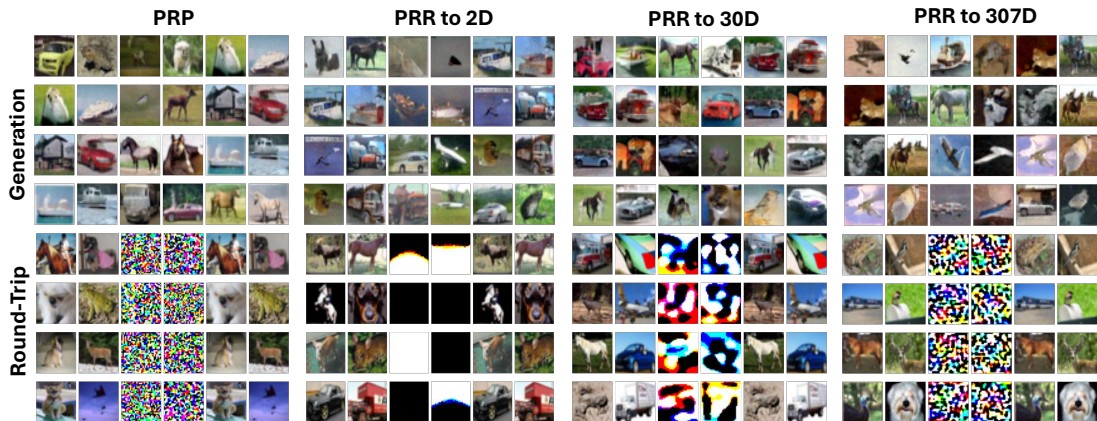

Figure 6: **Generation and Round-Trip Experiments for CIFAR-10 at IG=1.02 and varying number of preserved dimensions**. Layout and setup same as for **Figure 5** - see **Appendices B.5.2, B.5.1** for details.

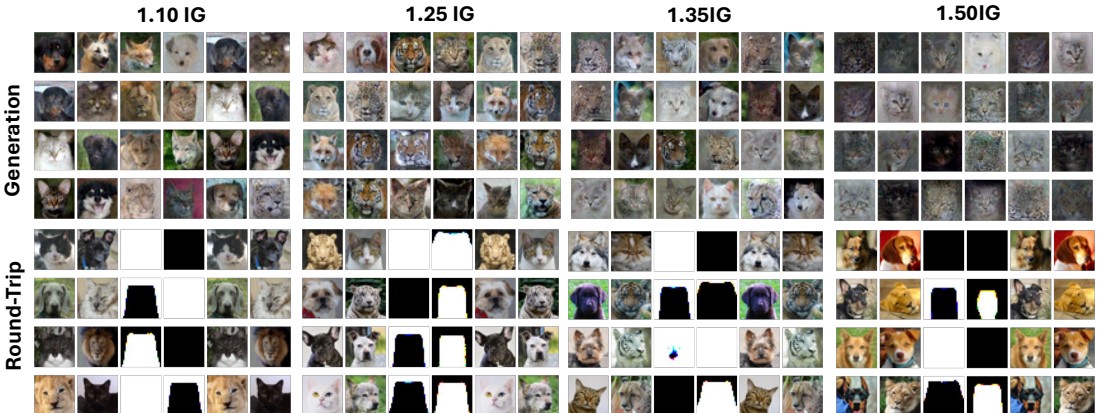

Figure 7: **Generation and Round-Trip Experiments for AFHQv2 dataset with dimension reduction to 2D (PRR to 2D) at different inflation gaps (IGs)**. **Top row:** Generated samples for each inflation gap (IG) flow schedule (1.10, 1.25, 1.35, and 1.50), all with $d = 2$. **Bottom row:** Results of round-trip experiments for same schedules. Leftmost columns are original samples, middle columns are samples mapped to Gaussian latent spaces, and rightmost columns are recovered samples.

### B.6 Details of Toy HMC Experiments

As highlighted in **Section 5**, we utilized Hamiltonian Monte Carlo (HMC) [1, 56–58] to assess if errors in our network score estimates could result in mis-calibrated posterior distributions. In these experiments, we worked with the toy 2D circles dataset (using both PR-Preserving and PR-Reducing schedules) and began by constructing our observed data samples $\mathbf{x_{obs}}$ as follows: **First**, we sampled a set of latent variables $\mathbf{z}$ from a 3-component Gaussian Mixture Model (GMM) $p(\mathbf{z}) = \sum_{i=0}^{2} w_i \mathcal{N}(\boldsymbol{\mu}_i, \boldsymbol{\Sigma}_i)$ with known means ($\boldsymbol{\mu}$), *diagonal* covariances ($\boldsymbol{\Sigma}$), and weights ($\mathbf{w}$) (**Table 14**). **Second**, we integrated the sampled $\mathbf{z}$ points backwards in time ("generation") using our proposed pfODEs with score estimates taken from trained networks to obtain "noise-free" observed data samples $\mathbf{x_{nl}}$. **Finally**, we added a small amount of isotropic Gaussian noise to these samples ($n \sim \mathcal{N}(0, \sigma^2)$, $\sigma^2 = 10^{-2}$), to obtain our final observed data, $\mathbf{x_{obs}}$.

Table 14: Ground-Truth Means, Covariance Diagonals, and Weights for Gaussian Mixture Model (GMM) Components Used in Toy HMC Experiments

| GMM Component | Schedule | Mean | Covariance Diagonal | Weight |
|---|---|---|---|---|
| $0^{th}$ | PR-Preserving | $[0.0, 0.0]$ | $[5.625 \times 10^{-1}, 5.625 \times 10^{-1}]$ | 0.50 |
| $0^{th}$ | PR-Reducing | $[0.0, 0.0]$ | $[5.625 \times 10^{-1}, 5.625 \times 10^{-3}]$ | 0.50 |
| $1^{st}$ | PR-Preserving | $[-5 \times 10^{-2}, 0.0]$ | $[10^{-2}, 1.0]$ | 0.25 |
| $1^{st}$ | PR-Reducing | $[-5 \times 10^{-2}, 0.0]$ | $[10^{-2}, 10^{-2}]$ | 0.25 |
| $2^{nd}$ | PR-Preserving | $[5 \times 10^{-2}, 0.0]$ | $[1.0, 10^{-2}]$ | 0.25 |
| $2^{nd}$ | PR-Reducing | $[5 \times 10^{-2}, 0.0]$ | $[1.0, 10^{-4}]$ | 0.25 |

We then used these observations, $\mathbf{x_{obs}}$, along with the HMC implementation provided in the `hamiltorch` library [56], to jointly sample from the posterior over $(\{\mathbf{z}_j\}, \mathbf{w})$, assuming $\{\boldsymbol{\mu}_i, \boldsymbol{\Sigma}_i\}$ known.

For both PR-Preserving and PR-Reducing experiments, we generated 2000 samples ($\mathbf{x_{obs}}$). For sampling, we used $L = 15$ steps per sampling trajectory, discarding the first 500 samples as "burn-in." Step sizes were $10^{-2}$ for PR-Preserving and $10^{-3}$ for PR-Reducing schedules. Because sampling required integration over the full generative trajectory and was slow to mix, requiring roughly 40 minutes per sample, we initialized our $\mathbf{w}$ and $\mathbf{z}_j$ estimates to ground truth values. In other experiments, we verified that other initializations quickly converged to these values, but this procedure avoided numerical instabilities associated with integration of the generative pfODE during the burn-in phase. Finally, to reduce sample autocorrelation, we thinned the resulting chains by a factor of 5.

As mentioned above, this procedure required multiple neural function evaluations (NFEs) for pfODE integration per HMC integration step, producing very long sampling times. For instance, using the single-GPU setup of `hamiltorch` required $\simeq 2$ weeks to pass burn-in for our PR-preserving schedule and $\simeq 4$ weeks for our PR-preserving schedule. As a result, sample numbers were somewhat small ($N = 1872$, PR-preserving; $N = 1635$, PR-reducing), and thinned traceplots still exhibited some considerable correlation (**Figure 8**), underscoring the impracticality of using sampling-based inference in these models.

## C  Appendix: Additional Experiments and Supplemental Information

### C.1  Spectra and PR-Dimensionality for a few common image datasets

Shown in **Table 15** are participation ratio (PR) values for some benchmark image datasets. **Figure 9** showcases spectra (zoomed in to first 25PCs) for same image benchmarks.

### C.2  Additional Toy Experiments

#### C.2.1  Toy Alpha-Shape/Mesh Coverage Experiments

To assess numerical error incurred when integrating our proposed pfODEs, we performed additional coverage experiments using 3D meshes and 2D alpha-shapes [91, 92] in select toy datasets (i.e., 2D

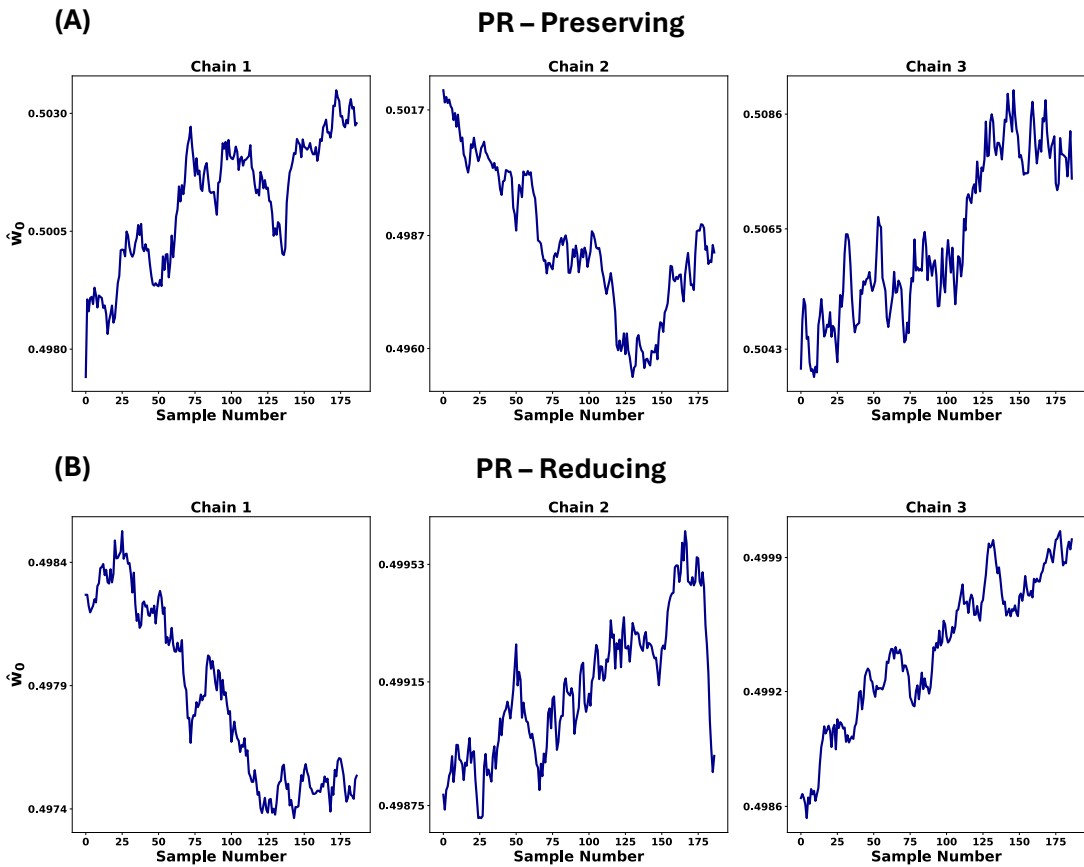

Figure 8: **Traceplots (post-thinning) for 3 random chains for PR-Preserving and PR-Reducing schedules**. **A:** Traceplots for 3 random PR-Preserving chains, after thinning by a factor of 5. "X axis" represents sample number and "Y axis" represents value of zeroth dimension of sample ($\hat{\mathbf{w}}_0$).**B:** Same set up, only for 3 random PR-Reducing chains. Note that there is still some considerable correlation in the samples, even after thinning. Additionally, mixing is *not* particularly good.

Table 15: Participation ratio (PR) for some commonly used image datasets.

| Dataset | PR |
|---|---|
| MNIST | 30.69 |
| Fashion MNIST | 7.90 |
| SVHN | 2.90 |
| CIFAR-10 | 9.24 |

circles and 3D S-curve), **Figure 10**. Here, we began by sampling 20K test points from a Gaussian latent space with appropriate diagonal covariance. For PR-Preserving schedules, this is simply a standard multivariate normal with either 2 or 3 dimensions. For PR-Reducing experiments, this diagonal covariance matrix contains 1's for dimensions being preserved and a smaller value ($10^{-2}$ for Circles, $2.5 \times 10^{-3}$ for S-curve) for dimensions being compressed.

Next, we sampled uniformly from the surfaces of balls centered at zero and with linearly spaced Mahalanobis radii ranging from 0.5 to 3.5 (200 pts per ball). We then fit either a 2D alpha-shape (2D Circles) or a mesh (3D SCurve) to each one of these sets of points. These points thus represent "boundaries" that we use to assess coverage prior to and after integrating our pfODEs. We define the initial coverage of the boundary to be the set of points (out of the original 20K test points) that lie inside the boundary. We then integrate the pfODE backward in time (the "generation" direction)

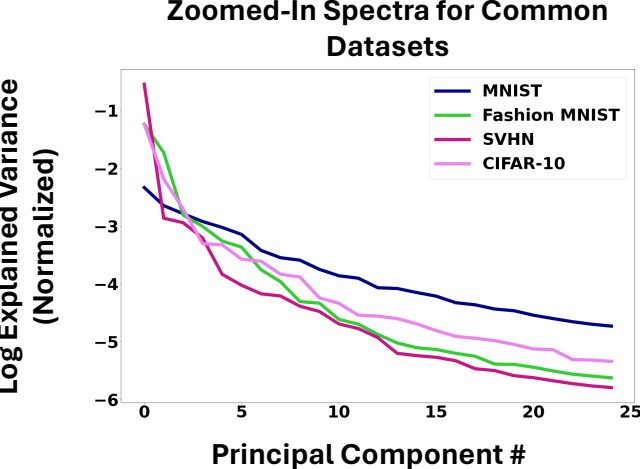

Figure 9: **Zoomed-in spectra for some standard image datasets.** Log of explained variance versus number of principal components (PCs) for 4 common image datasets (MNIST, Fashion MNIST, CIFAR-10, and SVHN). We plot only the first 25 PCs across all datasets to facilitate comparison.

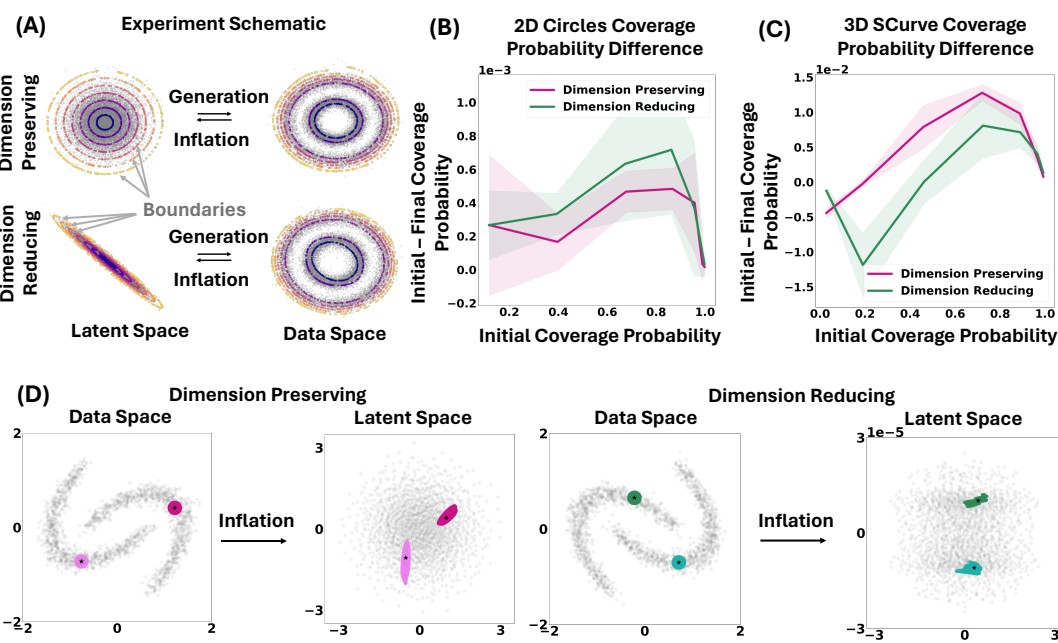

Figure 10: **Mesh/Alpha-Shape Calibration experiments.** For select toy datasets, we numerically assessed coverage during the inflation and generation procedures using (3D) meshes and (2D) alpha-shapes. **(A)** We constructed fixed coverage sets by sampling data points at fixed Mahalanobis radii from the centers of each distribution and creating alpha shapes (2D) or meshes (3D). **(B–C)** We then quantified the change in coverage fraction for each of these sets at the end of either "inflation" or "generation" procedures. Lines represent means and shaded regions ±2 standard deviations across three sets of random seeds. **(D)** Illustration of the effect of flows on set geometry. While both types of flows distort the shapes of initial sets, they do preserve local neighborhoods, even when one dimension is compressed by five orders of magnitude.

for each sample and boundary point. At the end of integration, we again calculate the mesh or 2D alpha-shape and assess the number of samples inside, yielding our final coverage numbers.

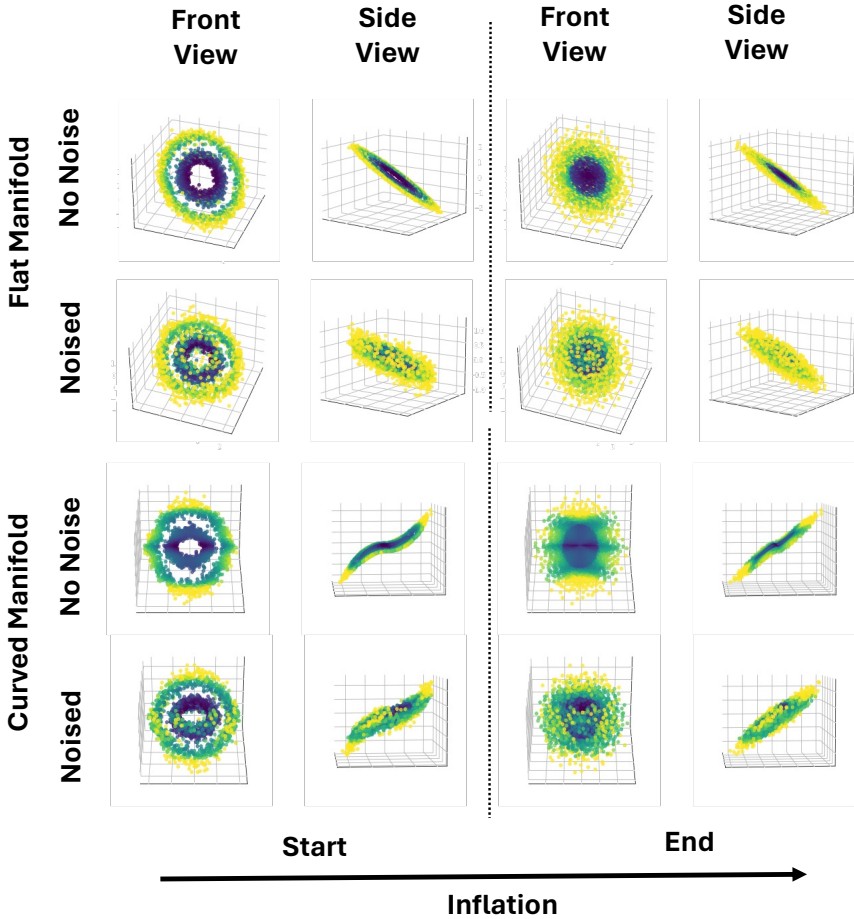

Figure 11: **Additional PR-Preserving experiments for 2D data embedded in 3D space.** Here we integrate our PR-Preserving pfODEs forwards in time (i.e., inflation) for 2 different toy datasets, constructed by embedding the 2D Circles data in 3 dimensional space as either a flat (top rows) or a curved (bottom rows) manifold. We present results for such simulations both without any added noise ($1^{st}$ and $3^{rd}$ rows) and with some small added noise (0.2 and 0.5 $\sigma$ for flat and curved cases, respectively - $2^{nd}$ and $4^{th}$ rows).

Similarly, we take our samples and boundary points at the end of generation, simulate our pfODEs forwards (i.e., the "inflation" direction), and once again, use 2D alpha-shapes and meshes to assess coverages at the end of this round-trip procedure. If our numerical integration were perfect, points initially inside these sets should remain inside at the end of integration; failure to do so indicates mis-calibration of the set's coverage. As shown in **Figure 10 B-C**), we are able to preserve coverage up to some small, controllable amount of error for both schedules and datasets using this process.

### C.2.2 Toy Experiments on Datasets with Lower Intrinsic Dimensionality

The pfODEs proposed here allow one to infer latent representations of data that either preserve or reduce intrinsic dimensionaltiy as measured by the participation ratio. In this context, it is important to characterize our PR-Preserving pfODEs' behavior in cases where data are embedded in a higher-dimensional space but are truly lower-dimensional (e.g., 2D data embedded in 3D space). In such cases, one would expect inflationary pfODEs to map data into a low-rank Gaussian that preserves the true intrinsic PR-dimensionality of the original data.

To confirm this intuition, we constructed 3D-embedded (2D) circles datasets using two different approaches: (1) by applying an orthonormal matrix $\mathbf{M}$ to the original data points, embedding them into 3D as a tilted plane (**Figure 11, top 2 rows**) or (2) constructing a third coordinate using

**Additional Toy $3D \rightarrow 2D$ Dimension-Reducing Simulations**

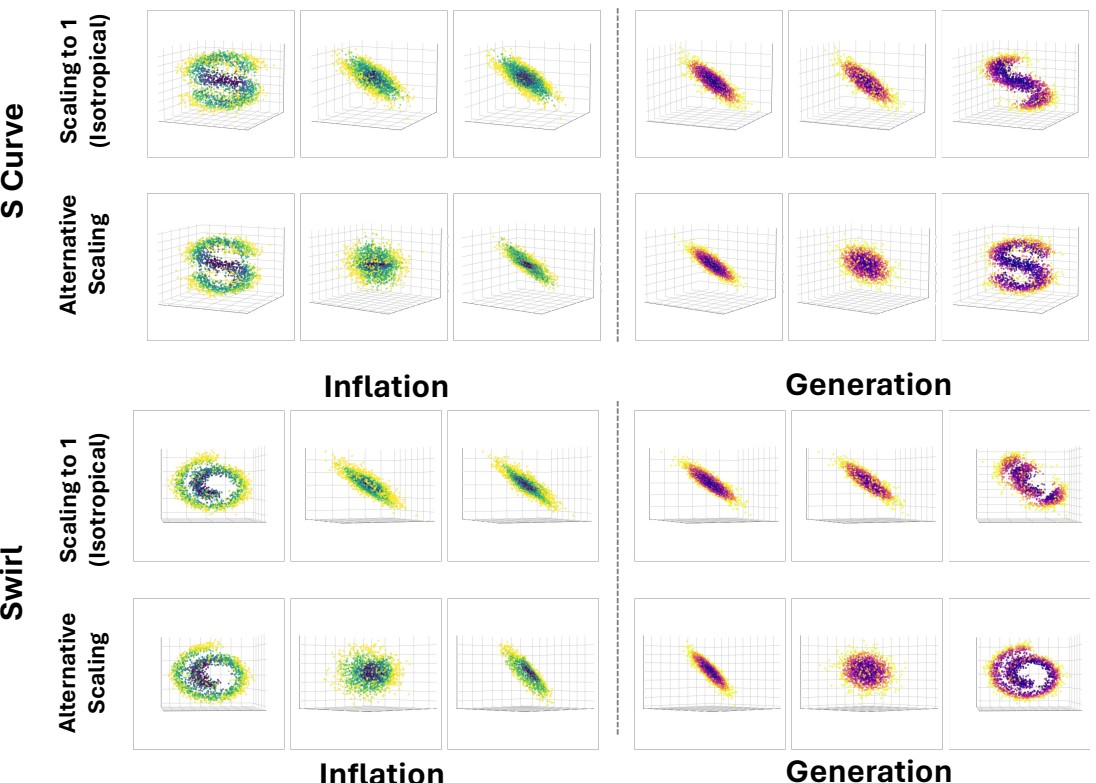

Figure 12: **Toy 3D → 2D dimension-reducing experiments with alternative scalings.** Shown here are simulations of our $3D \rightarrow 2D$ PR-Reducing pfODEs for 3D toy datasets (S-curve, Swirl) scaled either to unit variance across all 3 dimensions (first and third rows) or scaling the thickness dimension to 0.5, while leaving other dimensions scaled to 1 (second and fourth rows). Note that scaling all dimensions to 1 leads to some loss in original shape content when running generation (first and third rows, rightmost column). This is *not* the case when we make total variance contribution of the "thickness" dimension smaller (i.e., under the alternative scaling; second and fourth rows, rightmost column).

$z = \text{sign}(y)y^2$, which creates a curved (chair-like) shape in 3D (**Figure 11, bottom 2 rows**). We then simulated our PR-Preserving pfODE for both embedding procedures and considering both the case in which no noise was added to the data or, alternatively, where some Gaussian noise is added to the initial distribution, giving it a small thickness. We used zero-mean Gaussian noise with $\sigma$ of 0.2 and 0.5 for embedding types (1) and (2), respectively.

As shown in **Figure 11**, when no noise is added, our PR-Preserving pfODEs Gaussianize the original data points along the manifold plane (rows 1 and 3, rightmost columns). Alternatively, when noise is added and the manifold plane has some "thickness" the inflationary flows map original data into a lower-rank Gaussian (rows 3 and 4, rightmost columns). In both cases, the original PR is preserved (up to some small numerical error), as expected.

### C.2.3 3D Toy PR-Reducing Experiments with Different Dimension Scaling

For our 3D toy data PR-Reducing experiments, we tested how changing the relative scaling of different dimensions in the original datasets qualitatively changes generative performance.

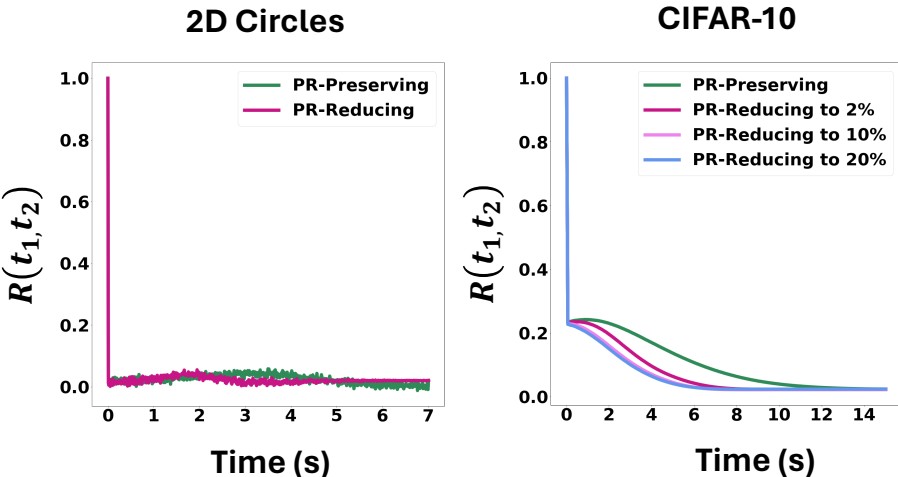

Figure 13: **Autocorrelation of denoiser network residuals.** Scaled autocorrelations of denoising network residuals $\epsilon(\mathbf{x}(t))$ for two sample toy networks (left, 2D circles PR-Preserving (green) and PR-Reducing to 1 dimension (pink)) and for networks trained on CIFAR-10 (right) for both PR-Preserving (green) and select PR-Reducing schedules (62D, $\approx 2\%$, (pink); 307D, $\approx 10\%$, (violet); 615D, $\approx 20\%$, (blue), all at IG=1.02). Toy data exhibit minimial autocorrelation along integration trajectories, while the CIFAR score estimates have some autocorrelation along one third to one half of the integration trajectory.

For the first experiment, we scaled all dimensions to variance 1 (**Figure 12, first and third rows**). In this case, all dimensions contribute equally to total variance in the data. In contrast, for the second experiment (**Figure 12, second and fourth rows**), we scaled the thickness dimension to variance 0.5 and all others to 1. In this case, the non-thickness dimensions together account for most of the total variance.

We then trained neural networks on 3D S-curve and Swirl data constructed using these two different scaling choices and used these networks to simulate our PR-Reducing pfODEs (reduction from $3D \rightarrow 2D$) both forwards (**Figure 12 left panels**) and backwards (**Figure 12 right panels**) in time. Of note, the first scaling choice leads to generated samples that seem to loose some of the original shape content of the target dataset (first and third rows, rightmost columns). In contrast, scaling choice 2 is able to almost perfectly recover the original shapes (second and fourth rows, rightmost columns). This is because scaling the thickness dimension to 0.5 reduces the percent of total variance explained along that axis, and our PR reduction preferentially compresses in that direction, preserving most information orthogonal to it. By contrast, the first scaling choice spreads variance equally across all dimensions and, therefore, shape and thickness content of target distribution are more evenly mixed among different eigendimensions. As a result, compressing the last dimension in this case inevitably leads to loss of both shape and thickness content, as observed here.

### C.3  Autocorrelation of Network Residuals

In **Section 5** above, we considered the possibility that numerical errors in approximating the score function might result in errors in pfODE integration and thus miscalibration of our proposed inference procedure. There, we argued that if these score estimation errors can be modeled as white noise, integration using sufficiently small integration step sizes will maintain accuracy, as dictated by theorems on numerical integration of SDEs [59]. Here, we investigate the validity of this approximation for our trained score functions.

As detailed in **Appendices B.1** and **B.3.1**, we did not directly estimate scores but trained networks to estimate a denoiser $\hat{\mathbf{y}} = \mathbf{D}_\theta(\mathbf{x}, \mathbf{C}(t))$, where $\mathbf{y}$ are samples from the data and $\mathbf{x} = \mathbf{y} + \mathbf{n}$ are the noised samples with $\mathbf{n} \sim \mathcal{N}(\mathbf{0}, \mathbf{C}(t))$. In this case, one can then compute scores for the noised

distributions using:

$$\nabla_{\mathbf{x}} \log p(\mathbf{x}, \mathbf{C}(t)) = \mathbf{C}^{-1}(t) \cdot (\mathbf{D}_\theta(\mathbf{x}, \mathbf{C}(t)) - \mathbf{x}) \tag{84}$$

In practice, however, this de-noised estimate contains some error $\epsilon = \hat{\mathbf{y}} - \mathbf{y}$, which is the true residual error in our network estimates. Therefore, we rewrite our score expression as:

$$\nabla_{\mathbf{x}} \log p(\mathbf{x}, \mathbf{C}(t)) = \mathbf{C}^{-1}(t) \cdot ((\hat{\mathbf{y}} - \mathbf{x}) + \epsilon) \tag{85}$$

where $(\hat{\mathbf{y}} - \mathbf{x})$ can be understood as the magnitude of the correction made by the denoiser at $\mathbf{x}$ [51]. Note that $\epsilon = \mathbf{0}$ for the ideal denoiser (based on the true score function), but nonzero $\epsilon$ will result in errors in our pfODE integration.

As argued above, these errors can be mitigated if they are uncorrelated across the data set, but this need not be true. To assess this in practice, we extracted estimation errors $\epsilon(\mathbf{x})$ across a large number of data samples (10K for 2D circles toys, 50K for CIFAR-10) and for networks trained on both PR-Preserving and select PR-Reducing schedules (PR-Reducing to 1D for circles at IG=2.0, and to 62D, 307D, and 615D for CIFAR-10, all at IG=1.02) and then computed cross-correlations for these errors along integration trajectories $\mathbf{x}(t)$:

$$\mathbf{R}(t_1, t_2) = \mathbb{E}_{\mathbf{x}}[(\epsilon(\mathbf{x}(t_1)) - \bar{\epsilon})(\epsilon(\mathbf{x}(t_2)) - \bar{\epsilon})^\top] \tag{86}$$

where $\bar{\epsilon}$ is the mean residual across the entire data set. In practice, we use scaled correlations in which an entry $R_{ij}$ is normalized by $\sigma_i\sigma_j$ the (zero-lag) variance of the residuals along the corresponding dimensions.

Results of these calculations are plotted in **Figure 13**, for the mean across diagonal elements of $\mathbf{R}$. As the left panel of **Figure 13** shows, residuals display negligible autocorrelation for networks trained to denoise toy data sets, while for CIFAR-10 (right panel), there is some cross-correlation at small time lags. This is likely due to the increased complexity of the denoising problem posed by a larger data set of natural images, in addition to the limited approximation capacity of the trained network. As a result, points nearby in data space make correlated denoising errors. Nevertheless, this small amount of autocorrelation does not seem to impact the accuracy of our round-trip experiments nor our ability to produce good-quality generated samples (**Figures 5, 6**; **Table 1**).

### C.4 Dataset Pre-Processing

Toy datasets were obtained from `scikit-learn` [93] and were de-meaned and standardized to unit variance prior to training models and running simulations. The only exceptions to this are the alternative 3D toy datasets detailed in **Appendix C.2.3**, where the third dimension was scaled to slightly smaller variance.

For CIFAR-10 and AFHQv2 datasets, we apply the same preprocessing steps and use the same augmentation settings as those proposed for CIFAR-10 in [49] (cf. **Appendix F.2**), with the only change that we downsample the original AFHQv2 data to $32 \times 32$ instead of $64 \times 64$.

### C.5 Licenses

Datasets:

- CIFAR-10 [61]: MIT license
- AFHQv2 [62]: Creative Commons BY-NC-SA 4.0 license
- Toys [93]: BSD License

