# OpenReview forum: "Inflationary Flows: Calibrated Bayesian Inference with Diffusion-Based Models"
_NeurIPS.cc/2024/Conference — NeurIPS 2024 poster_

### Official Review · Reviewer_13gp · 2024-06-18

**Soundness:** 2
**Presentation:** 2
**Contribution:** 2
**Rating:** 5
**Confidence:** 3

**Summary:**

This paper introduces inflationary flows, a diffusion model with deterministic denoising trajectories that allow mapping high-dimensional data to low-dimensional Normal distributions. The dimensionality reduction is controllable, and it is claimed that, as long as the score is correctly estimated, the method provides calibrated uncertainty estimates.

**Strengths:**

### Originality
* The method is new to me.

### Quality
* I did not identify flaws in the mathematical derivations but did not check derivations in the appendices carefully.

### Clarity
* The flow is good.
* Most of the implementation details are provided.
* The authors made an effort to be pedagogical and introduce all the necessary background.

### Significance
* Having calibrated uncertainty estimates is of high significance in risk-sensitive tasks.

**Weaknesses:**

### Originality
* No concerns about originality.

### Quality
* I have some concerns about the empirical evaluation of the method. My main concern is that there are no comparisons to existing methods while there exist plenty of generative models, such as other diffusion models, and normalizing flows,... Additionally, the experiments did not convince me that this method indeed provides calibrated uncertainty estimates, probably because I did not understand what Figure 4 shows (see clarity concerns).

### Clarity
* To me, the central part of the submission, the calibrated uncertainty estimates, is not detailed enough. Only 7 lines at the beginning of section 5 mention how to obtain a calibrated uncertainty estimate, and after reading the manuscript, it is still not clear to me as to why this method provides calibrated uncertainty estimates while others don't.
* I read Appendix B6 three times and still don't understand what is done, how Figure 4 is obtained, and what it is supposed to show.

### Significance
* This paper presents a method that produces calibrated estimates as long as the score is well estimated. My intuition would be that estimating the score correctly is the root of the problem. other methods, such as normalizing flows, also provide calibrated estimates if the neural network is perfectly trained. The miscalibration of other methods, given perfect score estimates, has not been quantified in order to showcase the significance of their method.
* It is not clear to me what this method brings in comparison to other diffusion models. The authors seem to claim that their method provides calibrated uncertainty estimates because it is based on a deterministic process, but as mentioned by the authors, equation (4) is also a deterministic process. What does this method bring that equation (4) doesn't have? Additionally, Normalizing flows are also deterministic transformations of data to a base (usually Normal) distribution and satisfy this property.  It is also not clear to me why a deterministic process is required for calibrated sampling-based uncertainty estimates (probably linked to clarity issues in section 5).

**Questions:**

* Do you have evidence that strong miscalibration can be observed when using other methods with perfectly learned scores that would strengthen the significance of your method?

* Is there any comparison available between your method and other methods both on the quality of the generated objects and on the calibration of the uncertainty estimates?

* Could you clarify exactly what properties this method has that no other existing methods have and how those properties lead to calibrated uncertainty estimates?

**Limitations:**

Limitations are properly addressed.

---

> ### Author Rebuttal · Authors · 2024-08-06
>
> We appreciate the author's time and effort in reviewing our paper and apologize for some confusion.
>
> Please see our General Rebuttal, in which we attempt to clarify several points raised by multiple reviewers, particularly
> - the particular calibration and modeling task we set out to solve
> - the difficulties of assessing posterior calibration in this task
>
> **Relation to other models:** Several of the reviewer's concerns related to the relationship of our approach to other models. Given our aim of providing calibrated posteriors over structured latent representations, we make the following comparisons:
>
> - **VAEs:** Variational autoencoders and related models offer structured latent spaces, but posterior uncertainties are approximated by Gaussians and often overconfident. Several approaches, including [1], attempt to mitigate this posterior mismatch. VAEs typically offer poor generative performance on challenging benchmark data sets.
> - **Simulation-based inference:** We are not experts on SBI, but our understanding is that these approaches usually assume a parameterized generative model (often likelihood-free and based on substantial domain knowledge) and aim for both good generative performance and well-calibrated inference over model parameters. By contrast, we do not assume a particular form for our generative model (more similar to VAEs and normalizing flows) and are focused on posterior uncertainty in latent variables, not generative model parameters.
> - **Diffusion models:** Diffusion models offer state-of-the art generative performance, and our model is trained as a diffusion model using a particular noise schedule. But standard diffusion models _do not_ produce well-organized latent representations (Figure 1, left). Rather, they are constrained to "latent" representations that are the same size as the data dimension, inflate the intrinsic data dimension, and mix widely disparate data together.
> - **Normalizing flows:** One reviewer pointed out that our introduction ignored a prominent line of work on injective normalizing flows, which map low-dimensional distributions to high-dimensional data [e.g., 2, 3, 4]. This is a regrettable omission on our part. Indeed, the goal in those works is very similar to our own. As with our model, a deterministic normalizing flow that changes dimensionality will, by definition, produce well-calibrated posteriors and low-dimensional latent representations. In fact, our model can be viewed as a species of continuous normalizing flow that works by learning an ODE rather than composing a sequence of functions. Our main advantage over these models is primarily in predictive calibration (i.e., generation quality), where our model inherits the performance of diffusion models in producing better coverage of the data distribution and much lower FID scores (cf. [5], FID scores for CIFAR-10 at $d=40$).
>
> Thus, VAE-like models and injective normalizing flows are the most suitable comparison model classes for our purposes.
>
> **Calculation of calibration:** We apologize for the confusion on this point, particularly the analysis underlying Figure 4. For reasons detailed in the General Rebuttal, it is extremely difficult to assess calibration for high-dimensional models, so our goal in this analysis was to assess the following question in the context of low-dimensional synthetic examples: **Given a set in the data space with a particular probability measure, can we identify a set in the latent space with the same probability measure that maps to it?**
>
> It is fairly obvious that if the score function is perfectly known and there are no numerical integration errors, this _must_ be the case: we just integrate the pfODE (6). However, if there are errors in estimating the score function or integrating the ODE, a set that contains 20% of points sampled from a probability distribution in $\mathbf{x}$ (data) space may only contain 10% of these points when transported to $\mathbf{z}$ (latent space).
>
> Figure 4 is our attempt to assess whether this is the case. We first trained a score function network to approximate the score for the datasets shown in (A). We then defined a collection of sets (dashed lines) at various Mahalanobis distances from the mean of the data distribution and calculated how much of the data was contained within each set. This is the "Initial Coverage Probability" used in (B) and (C). The next question is how to "transport" these sets to the latent space --- tricky, because they are defined as contours. To approximate this, we sampled boundary points on these contours, constructed meshes or alpha shapes based on them, and transported the sampled boundary points (and thus meshes) to the latent space. Finally, we counted the percentage of latent data points inside these sets. This is the "Final Coverage Probability." Figure 4B-C plot the differences in these two quantities, which should be 0 for a perfectly calibrated flow probability and are very small ($\mathcal{O}(10^{-3})$ or $\mathcal{O}(10^{-2})$). Again, this is expected mathematically because of the deterministic ODE; it is only intended as a check on the accuracy of the score function approximation and numerical integration.
>
> We hope this explanation clarifies things. If accepted, we plan to revise Section 5 and Appendix B.6 accordingly.
>
> [1] Rodriguez-Santana and Hernandez-Lobato. Adversarial α-divergence minimization for Bayesian approximate inference. Neurocomputing, 471:260–274, 2022.
>
> [2] Brehmer and Cranmer. Flows for simultaneous manifold learning and density estimation. NeurIPS 2020.
>
> [3] Caterini et al. Rectangular flows for manifold learning. NeurIPS 2021.
>
> [4] Cornish et al. Relaxing Bijectivity Constraints with Continuously Indexed Normalising Flows. ICML 2020.
>
> [5] Flouris and Konukoglu. Canonical normalizing flows for manifold learning. NeurIPS 2023.

---

> > ### Comment · Reviewer_13gp · 2024-08-12
> >
> > Thank you for all the clarifications. In light of those, I increase my score to 5. I look forward to reading the updated version with improved clarity and additional results regarding model comparisons mentioned in the general rebuttal!

---

> > > ### Author Response · Authors · 2024-08-12
> > >
> > > We appreciate the reviewer's suggestions and are working to implement these additional comparisons.

---

### Official Review · Reviewer_pn8r · 2024-07-12

**Soundness:** 3
**Presentation:** 3
**Contribution:** 3
**Rating:** 7
**Confidence:** 3

**Summary:**

The paper proposes inflationary flows (IFs) for sampling-based density estimation. IFs belong to the growing family of diffusion-based models (DBMs) that are trained to transform distributions via a continuous process. However, the current paper notes that typical DBMs actually increase the intrinsic dimensionality of the data in the process, which may be disadvantageous in certain cases. Thus the authors propose to constrain the time-varying covariance of the process in order to approximately preserve the intrinsic data dimensionality as indexed by the participation ratio metric. In addition, the constraint can be modified to encourage dimensionality reduction, with interesting implications for modeling nominally high-dimensional data. The properties and performance of IFs are evaluated across low-dimensional toy data sets and image benchmarks.

**Strengths:**

- The paper is well-written and the flow of argument can be easily followed. I particularly appreciate Section 2.

- The method is an interesting addition to the growing number of expressive sampling methods for density estimation, while allowing for preservation or reduction of the intrinsic data dimensionality.

- The method can be useful way beyond the empirical settings considered in the paper.

- The appendix is rich with useful details and additional results.

**Weaknesses:**

- Generic claims that NFs cannot operate on compressed representations and invalidated by recent work on relaxing architectural constraints [1-3]. These architectures need to be at least considered in the related work and possibly make it into the benchmarks as contenders.

- The abstract and introduction begin as if the paper is situated in the context of sampling methods for Bayesian estimation (i.e., conditional estimation), but doesn’t feature any examples relevant to the field (i.e., only unconditional density estimation) and only references generic generative models. Moreover, with this framing, it is written as if the field of simulation-based inference (SBI, [4]), or amortized Bayesian inference (ABI) in particular, does not exist. However, SBI has long been concerned with calibrated Bayesian inference (especially for non-linear, rather low-dimensional problems) and several recent papers have used continuous probability models, such as diffusion models [5] or consistency models [6], to name just a few.

- Section 3 can profit from a bit more consistency overall, for example some symbols change meaning, such as $\boldsymbol{\Sigma}_0$ and $\boldsymbol{\Sigma}$, P4L141 seems to imply that $\boldsymbol{\Sigma}(t)$ is diagonal for any $t$, but in my understanding this will not hold for $t \approx 0$, PR is defined as a function over matrices (7) and vectors (8), etc.

- An empirical example where the dimensionality-preserving property of IFs actually brings about a palpable advantage over a standard DBM would further strengthen the paper.

**Questions:**

- Does the model admit a procedure for density estimation in addition to sampling? Is there a danger of this density ending up degenerate due to errors in compression?

**Limitations:**

The authors openly discuss limitations of their method and future work.

---

> ### Author Rebuttal · Authors · 2024-08-06
>
> We want to thank the reviewer for their positive assessment of our work. We believe they have done a good job of articulating our contributions.
>
> In responses the weaknesses noted by the reviewer:
>
> - We regret our omission of injective normalizing flows (e.g., [1-4]) from the introduction and related work sections and thank the reviewer for bringing them to our attention. These models have very similar goals to our own, and of course, our approach can be viewed as a species of continuous normalizing flow (see discussion of Flow Matching in the General Rebuttal). Like ours, these models should have well-calibrated posteriors to the degree that they are deterministic, though we note that the connection with Bayesian inference is _not_ often discussed in the flow matching literature. Although we could not perform comparison experiments against these models during the rebuttal period, we believe, based on previously reported FID scores for benchmark image data (see Table 1 of [4], CIFAR-10), that our model would have better predictive calibration (i.e., better FID scores). If accepted, we also plan to include comparisons with several of these normalizing flow models in the simplified inferential setup described in the General Rebuttal, where we can compare model inferences with MCMC ground truth.
>
> - We apologize for the confusion and have attempted to clarify these points in our General Rebuttal and our reply to Reviewer 13gp, but to reiterate briefly:
>     - We are focused in this work on the problem of unconditional generative modeling in the absence of an assumed likelihood model. That is, as with VAEs, we are interested in both learning a generative model and a latent space at the same time. We are big fans of simulation-based inference, but our understanding is that these approaches typically begin with a well-defined generative model and then attempt to perform approximate inference on the parameters of this model. By contrast, we do not assume a fixed parameterization or generative model, which creates issues when assessing calibration of inferences.
>
>      However, we agree that it is important to explain this distinction, which we would plan to add to the introduction of a revised version if accepted.
>
>     - As also discussed in the General Rebuttal, our primary focus in this work is on calibration of inferences, not predictions. While both are important, we have focused on the former, though as detailed in our rebuttal to Reviewer v9FZ and in the attached pdf, we have performed additional predictive calibration experiments.
>
> - We have checked, and we believe the notation in Section 3 is consistent. $\boldsymbol{\Sigma} = \boldsymbol{\Sigma}(t) = \boldsymbol{\Sigma}_0 + \mathbf{C}(t)$ is the time-dependent covariance of the smoothed data, where $\boldsymbol{\Sigma}(0) = \boldsymbol{\Sigma}_0$ is the covariance of the data. Moreover (ll.142-143), we choose $\mathbf{C}(t)$ to be diagonal in the eigenbasis of the data, $\boldsymbol{\Sigma}_0$, so that $\boldsymbol{\Sigma}(t)$ does indeed remain diagonal for all $t$. It's for this reason, along with the fact that PR is invariant to linear transformations (ll. 136-137) that the definition in (8) is the same as the one in (7). That is, in the eigenbasis of $\boldsymbol{\Sigma}_0$, $\boldsymbol{\Sigma}(t) = \mathrm{diag}(\boldsymbol{\sigma^2}(t))$.
> - We view our method as distinct in aim from DBMs. While our flows can be used as generative models, data compression does result in FID scores slightly worse than those for the best DBMs. What inflationary flows provide that DBMs do not are neighborhood-preserving, low-dimensional latent spaces.
> - This is an interesting question. One might think of integrating the Fokker-Planck Equation (3) backward from the smoothed low-rank Gaussian distribution, but because _both_ the compression and smoothing eliminate information, there is an exponential loss of initial conditions, and integrating backward from a Gaussian (i.e., the asymptotic solution) cannot recover the data. We think the problem of density estimation would be interesting to pursue here but don't currently have a solution to it. This is an advantage offered by some manifold learning normalizing flows over our approach.
>
> [1] Brehmer and Cranmer. Flows for simultaneous manifold learning and density estimation. NeurIPS 2020.
>
> [2] Caterini et al. Rectangular flows for manifold learning. NeurIPS 2021.
>
> [3] Cornish et al. Relaxing Bijectivity Constraints with Continuously Indexed Normalising Flows. ICML 2020.
>
> [4] Flouris and Konukoglu. Canonical normalizing flows for manifold learning. NeurIPS 2023.

---

> ### Comment · Reviewer_pn8r · 2024-08-09
> **Reviewer Response**
>
> I thank the authors for their professional response and the clarifications. I also noticed that my references went missing in my original review, so I re-post them below. Having also read the other reviews, I will increase my score and would really like to see the comparisons with non-bijective flows.
>
>
> [1] Draxler, F., Sorrenson, P., Zimmermann, L., Rousselot, A., & Köthe, U. (2024). Free-form flows: Make any architecture a normalizing flow. In International Conference on Artificial Intelligence and Statistics (pp. 2197-2205). PMLR.
>
> [2] Sorrenson, P., Draxler, F., Rousselot, A., Hummerich, S., Zimmermann, L., & Köthe, U. (2024). Lifting architectural constraints of injective flows. In The Twelfth International Conference on Learning Representations.
>
> [3] Kothari, K., Khorashadizadeh, A., de Hoop, M., & Dokmanić, I. (2021, December). Trumpets: Injective flows for inference and inverse problems. In Uncertainty in Artificial Intelligence (pp. 1269-1278). PMLR.
>
> [4] Cranmer, K., Brehmer, J., & Louppe, G. (2020). The frontier of simulation-based inference. Proceedings of the National Academy of Sciences, 117(48), 30055-30062.
>
> [5] Sharrock, L., Simons, J., Liu, S., & Beaumont, M. (2022). Sequential neural score estimation: Likelihood-free inference with conditional score based diffusion models. arXiv preprint arXiv:2210.04872.
>
> [6] Schmitt, M., Pratz, V., Köthe, U., Bürkner, P. C., & Radev, S. T. (2023). Consistency Models for Scalable and Fast Simulation-Based Inference. arXiv preprint arXiv:2312.05440.

---

> > ### Author Response · Authors · 2024-08-12
> >
> > We appreciate the reviewer providing these references. We did find several of these but missed some others. We are likewise eager to see the outcome of the injective flows comparisons.

---

### Official Review · Reviewer_v9FZ · 2024-07-12

**Soundness:** 4
**Presentation:** 4
**Contribution:** 3
**Rating:** 7
**Confidence:** 4

**Summary:**

This paper presents a novel approach to Bayesian inference using diffusion-based models (DBMs). Traditional Bayesian inference methods struggle with high-dimensional integrals, and existing approximation methods either scale poorly (sampling-based) or offer limited theoretical guarantees (variational methods). This work leverages DBMs to develop "inflationary flows": these flows utilize a deterministic map from high-dimensional data to a lower-dimensional Gaussian distribution, achieved through ODE integration. The proposed method ensures local neighborhood preservation and controllable numerical error, enabling accurate uncertainty quantification. The approach also includes novel noise schedules and demonstrates high generative performance, even under significant data compression.

**Strengths:**

* The article is very well written and structured. Moreover, the authors provide good explanations that aid in developing intuition around the model, s.a. the diagram presented in Figure 1.
* The idea behind the inflationary flows is a very interesting one from the point of view of the properties of the method. It does seem to provide new key features that are missing in the literature and that may be helpful in many other problems.
* The theoretical contributions made by the authors are strong, with enough support behind the main points. The discussion makes it so that the inflationary flows model seems a natural consequence of previous works, filling previous gaps in the literature.
* The method produces calibrated, identifiable Bayesian inference by leveraging a deterministic map between the original data and a Gaussian distribution. This ensures accurate propagation of uncertainties from the data to the latent space, leading to reliable predictive distributions.

**Weaknesses:**

* While it is true that the KL divergence used in VI induces a mode-seeking behavior, this can be easily fixed alongside the properties of the posterior distribution obtained via VI. This is explored in articles s.a. [1], although it has become a widely-known fact for some time [2] and its properties are usually exploited in the context of VI-based methods. $\alpha$-divergences improve the performance of VI-based methods with very simple changes in the formulation. I think this should be discussed in the context of the introduction and the related work of the article.
* The accuracy of the method heavily depends on the quality of the estimated score function. Errors in this estimation can propagate through the model and impact the final results.
* The training process for diffusion-based models requires significant computational resources, particularly due to the need for training over larger noise ranges. This can limit the practical applicability of the method to very large datasets.
* I would appreciate it if the authors provided further metrics to fully ensure the quality of the predictive distributions. These can range from the usual log-likelihood estimates to the CRPS or other proper scoring rules [3], alonside the ECE compared to other baseline models (e.g. some based on VI, regular DBMs and MCMC-based approaches). This would help in comparatively assessing the quality of the predictions obtained in a broader sense.

## Minor:
* Check again for typos, e.g. "asympototically" (first paragraph of the introduction). I only found this one, but it is worth checking again.
* On the first equation, add the definition of $\mathbf{W}$ for completeness (as is done later for the time-reversed $\bar{\mathbf{W}}$).
* Although I do like the representation, I would add some short descriptions on the arrows of Figure 1 connecting the three paradigms (diffusion model $\rightarrow$ Fokker-Plank $\leftarrow$ pfODE), making it more explicit.

_(please see the 'Limitations' section for the references)_

**Questions:**

* Could you please discuss the potential impact of errors in the score function estimation on the accuracy and stability of the pfODE integration? How do you address the sensitivity of your method to these errors, and what strategies can be employed to mitigate their effects? This would be particularly interesting for complex datasets.
* When is it the PR measure for dimensionality expected to perform best? Are there any cases where this measure can lead to a bad performance due to the reliance only on second-order statistics?

**Limitations:**

* The article relies on the participation ratio (PR) as a measure of dimensionality, which might not capture more complex data structures effectively.
* Accuracy of the method depends heavily on the quality of the estimated score function, which can introduce errors if not precisely learned.
* The need for training diffusion-based models over larger ranges of noise can lead to trade-offs in compression performance and score estimation accuracy.
* The experimental phase can be bolstered with the addition of a few extra comparisons with other VI-based models, as well as regular DBMs and MCMC-based models in order to provide a fuller comparative picture of the performance of the proposal.

## References:
[1] Rodríguez Santana, S., & Hernández-Lobato, D. (2022). Adversarial $\alpha$-divergence minimization for Bayesian approximate inference. Neurocomputing, 471, 260-274.

[2] Minka, T. (2005). Divergence measures and message passing (p. 17). Technical report, Microsoft Research.

[3] Gneiting, T., & Raftery, A. E. (2007). Strictly proper scoring rules, prediction, and estimation. Journal of the American statistical Association, 102(477), 359-378.

---

> ### Author Rebuttal · Authors · 2024-08-06
>
> First, we want to thank the reviewer for their careful assessment. We believe the reviewer has correctly identified the aims and strengths of our work.
>
> As to the weaknesses:
>
> - The reviewer is correct that several alternative variational approaches have been able to mitigate some of the shortcomings of vanilla VAEs for predictive accuracy and uncertainty estimation. We are happy to include a discussion of these efforts in our introduction.
>
>  However, we will note that works such as [1] focus on calibration of _predictions_ rather than of _inferences_. While both are important, our focus here is on the latter (see our General Rebuttal).
>
>  Thus, in the attached pdf, we include results from additional predictive calibration experiments on CIFAR-10 using 3 different types of VAE models as comparisons: a vanilla VAE (i.e., reverse-KL), maximum mean discrepancy (MMD) VAE, and an $\alpha$-VAE [1], which uses the dual skew-geometric Jensen-Shannon (gJS) divergence with a learnable skew parameter. For these experiments, we utilized a similar implementation to the one used in [2] and trained models with latent spaces equal (in size) to the ones for our 2, 10, and 20% PR-Reducing experiments. All models were trained for 1K epochs, with $1e^{-4}$ learning rate, 256 batch-size, and the same architecture as detailed on Table 3 of [2] (see architecture for MNIST; we varied only latent space sizes to match compression levels in our experiments). As the table shows, these models all produce substantially worse FID scores than our approach, even when given the easier task of simply reconstructing data, no _de novo_ generation.
>
> - We agree with the reviewer that because our model is deterministic, it preserves neighborhoods by construction, with the result that accuracy in score function estimation is the central difficulty in practice. In response, we note make three observations in our work that speak to this concern:
>     - As argued in the main text (ll. 206-210), when score estimates are accurate in the mean, high variance in score estimates can be mitigated by using smaller integration step sizes.
>     - As we demonstrate in Appendix C.3 (cf. Figure 10), score function estimates do exhibit autocorrelation across flow time in the case of our larger data sets. This is less than ideal, since it means that fluctuations in the estimates may accumulate during integration, even with smaller step sizes. This is inherited from the methods we use to train the corresponding diffusion models and might be mitigated with better DBM training methods.
>     - Nonetheless, even when the score function is estimated inaccurately, if the final marginal distribution is a multivariate Gaussian, the model has learned a well-behaved latent space. That is, local inaccuracies in score approximation may be benign from the standpoint of learned latent spaces. In the rebuttal pdf, we have included q-q plots for both the first and last three principal component dimensions for some of our AFHQv2 experiments (both dimension-preserving and dimension-reducing to 2%), demonstrating that, regardless of possible score errors and autocorrelation, the final learned latent representation is quite close to Gaussian.
>     - Agreed. Diffusion models indeed require significant training resources, as our experiments show. However, as detailed in our General Rebuttal, our recent experiments with flow-matching training methods have significantly reduced these requirements, down to minutes rather than hours on toy data, increasing our confidence that methods like ours may be trained efficiently enough to compete with less intensive methods like VAEs.
>     - While, as we noted above, our primary focus in this paper is on the calibration of inferences (correct $p(\mathbf{z}|\mathbf{x})$), we agree that it is also important that the generation process be well calibrated (correct $p(\mathbf{x}) = \int p(\mathbf{x}|\mathbf{z}) \pi(\mathbf{z}) \, d\mathbf{z}$). To this end, we did additional predictive calibration experiments using VAEs (as detailed above) and plan to also include similar experiments using injective normalizing flows in a revised version.
>     - We expect the participation ratio to perform well when the data we wish to preserve are well aligned with the dimensions of highest variability, which will usually be cases where manifold curvature is mild. In Appendix C.2.1 (cf. Figure 8), for instance, we found that PR-reducing flows did not "stretch" the curved 2d manifold so much as "squish" it. We believe that the problem of identifying other intrinsic dimensionality measures that can be used to derive improved flows is a valuable direction for future research.
>
>
> [1] Rodriguez-Santana and Hernandez-Lobato. Adversarial α-divergence minimization for Bayesian approximate inference. Neurocomputing, 471:260–274, 2022.
>
> [2] Deasy et al. Constraining variational inference with geometric Jensen-Shannon Divergence. NeurIPS 2020.

---

### Official Review · Reviewer_y6tA · 2024-07-12

**Soundness:** 4
**Presentation:** 4
**Contribution:** 4
**Rating:** 6
**Confidence:** 4

**Summary:**

This paper proposes a new version of a probability flow-based diffusion model that allows for the effective reduction in dimensionality of the data after it has been diffused. Furthermore, the proposed approach allows for proper uncertainty measures of the data by preserving local neighborhoods from latent space to data space---in both dimension preserving and dimension reducing setups. Extensive empirical results are generated demonstrating the effects of the proposed scheme.

**Strengths:**

This is a unique and interesting problem to try and solve with diffusion models. The approach is well justified and presented clearly. The experimental setups are well motivated and the corresponding results are largely convincing.

**Weaknesses:**

For the experiments with high-dimensional data, I found the trend of having a higher percentage of dimension preservation resulting in worse performance to be concerning and the lack of investigation on this to be unsatisfactory. Only values of 2%, 10%, and 20% were chosen with the explanation of the results being "retaining more dimensions in our PR-Reducing schedules leads to larger scale gaps between our preserved and compressed dimensions (i.e., larger $e^{ρ(g_{*}−g_i)T}$), thus increasing the required noise range over which networks must optimally estimate scores;"  however, I do not believe that this tells the full story. The dimension-reducing flow approaches the original dimension-preserving flow as the number of $K$ reduced dimensions approaches the original data dimensionality $N$. As such, one would expect the performance to at some point start improving again. Should it not, and experience a discontinuity in performance from $K=N-1$ to $K=N$ then that would be interesting to note. Additionally, clearly it stands to reason that we cannot keep reducing the dimensionality further and further without eventually experiencing a decrease in performance, i.e., $K=1$ is most likely going to perform poorly for images. Finding out where this inflection point would be interesting to note as that is naturally going to be the optimal choice.

With the lack of comparison to other methods, which is understandable in and of itself given the focus of the paper, it is even more important then to fully explore the failure modes and general behavior of the proposed method.

**Questions:**

No additional questions, please address the main weakness I brought up. Please let me know if I misunderstood something as well. I am more than happy to be convinced otherwise on my points.

**Limitations:**

The authors adequately discussed the limitations.

---

> ### Author Rebuttal · Authors · 2024-08-06
>
> We appreciate the reviewer's positive assessment of our work. We hope that our replies to other reviewers, as well as the General Rebuttal, reinforce this assessment.
>
> The reviewer also raises an important issue in our experiments with high-dimensional image data: the paradoxical finding that performance appears to be _worse_ as _more_ dimensions are retained the PR-reducing flows. As we attempted to explain in Appendix B.2, this is a direct consequence of our choice of $\mathbf{g}$, the variable that determines our inflation schedule. In the experiments underlying Table 1, we chose $g_i = 2$ for preserved dimensions and $g_i = 1 - \frac{d}{D - d}$ for compressed dimensions (with $d$ the preserved and $D$ the number of data dimensions), so that $\sum_i g_i = D$. We also fixed the total inflation time $T$ and inflation rate $\rho$ across experiments. Given these choices, the quantity we call the inflation gap, the ratio between the variances of the preserved and compressed dimensions, is given by $e^{\rho(g_{\mathrm{max}} - g_{\mathrm{min}})T} = e^{\rho T D/(D - d)}$, and this is \emph{larger} when $d$ is higher (cf. Table 5 on p26, last 2 rows). That is, when we elect to preserve more dimensions, we perform more compression  (cf. Table 7 on p27), which, as explained in the main text, results in a much more difficult learning problem, since scores must be estimated over this larger range of scales.
>
> We hypothesize that, were we to choose $\mathbf{g}$, $T$, and $\rho$ to hold the inflation gap constant as $d$ is varied, preserving more dimensions should indeed result in better FID scores, and we plan to include these additional experiments (which the rebuttal period does not allow time for performing) in a revised manuscript.

---

> > ### Comment · Reviewer_y6tA · 2024-08-10
> >
> > Thank you for your response. Without the additional experiments investigating the full spectrum of behavior as $d$ varies from $1$ to $D$, I will have to maintain my original score with the hopes that when this paper is accepted (either here or at another venue) that these experiments are included to tell the full store.

---

> > > ### Author Response · Authors · 2024-08-12
> > >
> > > Thanks again for your helpful comments. We are currently running these experiments and will report back if they are completed before the end of the discussion period.

---

### Author Rebuttal · Authors · 2024-08-06

First, we appreciate the reviewers' thoughtfulness in assessing our paper.  While all reviewers agreed that the work is novel, they also noted weaknesses and requested clarification on several shared themes:

## Problem setup

**First,** we are concerned here with _unconditional generation_ in cases where **the generative model is unknown**. This is the most common problem setup in work on, e.g., VAEs, normalizing flows, and GANs, and is the one we adopt here. While conditional generation is an important problem, we restrict ourselves here to sampling data $\mathbf{x}$ based on some latent variables $\mathbf{z}$.

**Second,** we are interested in learning models with **structured latent representations.** This is typical of both VAEs and injective flows but not of GANs and diffusion models. As Figure 1 (left) illustrates, the diffusion procedure necessarily mixes data in the latent space, resulting in an unstructured latent representation. Thus, _we aim to construct models that combine the latent representations of NFs or VAEs and the generative power of DBMs._

**Finally,** we are interested in the **calibration of inferences.** We particularly apologize for confusion on this point, since it appears to be at the root of several questions posed by reviewers. To be more specific: in the models we consider, one could speak of both _predictive_ calibration --- how well the marginal $p(\mathbf{x})$ matches the data --- and _posterior_ calibration --- how the model posterior $q(\mathbf{z}|\mathbf{x})$ matches the true posterior $p(\mathbf{z}|\mathbf{x})$. Our primary focus is on the latter, though we will have more to say about the former below.

## Other approaches and our contribution
Please see our detailed reply to Reviewer 13gp.

## Model comparisons
All reviewers asked for more quantitative comparisons with other models.

**First,** for predictive calibration, we can easily calculate measures like FID and, e.g., the maximum mean discrepancy (MMD) between samples from our model and training data. In Table 1 of the attached pdf, we report FID scores for $\alpha$-VAEs, MMD-VAEs, and vanilla VAEs trained on CIFAR-10. Since VAEs are known to have low generative performance, it is no surprise that our model outperforms these by a wide margin. While injective flow methods also require substantial training time (cf. [5], Table 4), we plan to include these additional comparisons in a revised version.

**Second,** For posterior calibration, the situation is tricky. For cases where the generative model is known and parameters are shared across the data, MCMC methods can yield samples from the posterior for low-dimensional models. But for methods like ours, where the generative model is learned, the latent spaces are not necessarily comparable across models, and so "ground truth" is not shared. That is, the true $p(\mathbf{z}|\mathbf{x})$ is conditioned on the generative model, and the latent variables are distinct for each data point.

In this case, apart from the geometric approach we have taken in Figure 4 (see clarification in our rebuttal to Reviewer 13gp), the best method we have found for assessing calibration comes from [2], which generated $\mathbf{x}$s from a parameterized prior over $\mathbf{z}$ and then performed inference on the parameters of this prior using both their model and MCMC. Again, because of the deterministic nature of normalizing flows and our model, we expect both to perform very well on this task, with VAEs faring somewhat worse. If accepted, we plan to include this comparison for a selection of models from both classes.

## Connection with Flow Matching
Finally, since the initial submission of this article, we have established that our method can be linked to recent flow-matching approaches [e.g., 6, 7, 8]. Essentially, our DBM noising schedule can be recast as an "interpolant" in the terminology of [8], meaning that our proposal is essentially a novel, dimension-reducing form of flow-matching.

More importantly, experiments with our toy data examples (Figure 2 of the attached pdf) have found that training using flow matching is not only feasible, but also significantly speeds up learning --- as much as _two orders of magnitude_. This suggests that models like ours could be learned using drastically fewer resources than reported in our initial submission, a direction we intend to continue exploring.

[1] Flouris and Konukoglu. Canonical normalizing flows for manifold learning. NeurIPS 2023.

[2] Brehmer and Cranmer. Flows for simultaneous manifold learning and density estimation. NeurIPS 2020.

[3] Lipman et al. Flow matching for generative modeling. ICLR 2023.

[4] Liu et al. Flow straight and fast: Learning to generate and transfer data with rectified flow. ICLR 2023.

[5] Albergo and Vanden-Eijnden. Building normalizing flows with stochastic interpolants. ICLR 2023.

---

### Decision · Program_Chairs · 2024-09-25

**Decision:**

Accept (poster)

**Comment:**

This paper presents a new class of generative models and associated inference algorithm that maps high-dimensional data to a lower-dimensional Gaussian via ODE integration. The main claimed contribution of the paper is that the method is well calibrated and identifiable, i.e., is uniquely defined on the low-dimensional space.

Overall, the reviewers agree that the proposed approach is novel but raised several important concerns including: (1) Lack of comparisons with other approaches (including injective normalizing flows); (2) clarity regarding evaluation of calibrated uncertainties; and (3)  somewhat unintuitive results when more dimensions are retained (as the metrics get worse). The authors have, to some extent, addressed these issues satisfactorily in their comments by clarifying where their approach lies (unconditional generative modeling and learning structured representations) and provided additional results. However, the comparison with injective flows was not done due to time constraints, so I’d expect to see this in the final version of the paper. The authors have also clarified the relation of their work to flow matching by stating that it is essentially a “novel, dimension-reduction form of flow-matching”.

One additional point I’d like to see discussed in the final version is concerned with calibration. Usually in the literature this refers to predictive calibration. However, how well the approximate posterior matches the true posterior is simply an evaluation of the posterior approximation method. While posterior comparisons across different models does not make sense, one can still evaluate (predictive) calibration, which the authors have done. I also agree that uncovering structured latent representations is an important problem, but in my view this has more to do with the underlying model than with the approximation method (while I recognise these two may be conflated sometimes). I think more clarity in the terminology and evaluation should be provided.

 Given the positive comments and the recognition of the novelty of this work, I recommend acceptance.